
**Genesis of Diamond Dust and Thick Cloud Episodes observed**
**above Dome C, Antarctica**
**Philippe Ricaud[1], Eric Bazile[1], Massimo del Guasta[2], Christian Lanconelli[3,4], Paolo**
**Grigioni[5], Achraf Mahjoub[1]**
[1]Météo-France/CNRM, CNRS UMR 3589, 42 avenue Gaspard Coriolis, 31057 Toulouse,
France
[2]CNR, Via Madonna del Piano 10, 50019, Sesto Fiorentino, Italy
[3]Institute of Atmospheric Sciences and Climate (ISAC), Consiglio Nazionale delle Ricerche,
via Gobetti 101, 40129 Bologna, Italy;
[4]Now at Joint Research Center, Institute for Environment and Sustainability (IES), Land
Resource Management Unit (H05), via Fermi, 21027 Ispra (VA), Italy
[5]ENEA, Lungotevere Thaon di Revel, 76-00196 Roma, Italy
**Version V7, 15 September 2016**
**To be submitted to:** *Atmospheric Chemistry and Physics Discussions*





**Abstract**
From 15 March to 8 April 2011 and from 4 to 5 March 2013, the atmosphere above Dome
C (Concordia station, Antarctica, 75°06'S, 123°21'E, 3233 m amsl) has been probed by
several instruments and model to study episodes of thick cloud and diamond dust (cloud
constituted of suspended ice crystals). 1) A ground-based microwave radiometer
(HAMSTRAD, $H_2O$ Antarctica Microwave Stratospheric and Tropospheric Radiometers)
installed at Dome C that provided vertical profiles of tropospheric temperature and absolute
humidity to calculate Integrated Water Vapour (IWV). 2) Daily radiosoundings launched at
12:00 UTC at Dome C. 3) A tropospheric aerosol Lidar that provides aerosol depolarization
ratio along the vertical at Dome C. 4) Down- and upward short- and longwave radiations as
provided by the Baseline Surface Radiation Network (BSRN) facilities. 5) Space-borne
aerosol depolarization ratio from the Cloud-Aerosol Lidar with Orthogonal Polarization
(CALIOP) Lidar aboard the Cloud-Aerosol Lidar and Infrared Pathfinder Satellite
Observation (CALIPSO) platform along orbits close to the Dome C station. The time
evolution of the atmosphere has also been evaluated by considering the outputs from the
meso-scale AROME and the global-scale ARPEGE meteorological models. Two distinct
periods are highlighted by all the datasets: the warm and wet periods (24-26 March 2011 and
4 March 2013) and the cold and dry periods (5 April 2011 and 5 March 2013). Combining
radiation and active measurements of aerosols with nebulosity calculations, a thick cloud is
detected during the warm and wet periods with high depolarization ratios (greater than 30%)
from the surface to 5-7 km altitude associated with precipitation of ice particles and the
presence of a supercooled liquid water (depolarization of about 10%) cloud. During the cold
and dry periods, high depolarization ratios (greater than 30%) to a maximum altitude of 100-
500 m are measured suggesting that the cloud is constituted of ice crystals with no trace of
precipitation. These ice crystals in suspension in the air are named diamond dust. Considering



5-day back trajectories from Dome C and global distributions of IWV over the Antarctic show
that the thick-cloud episode is attributed to air masses with an oceanic origin whilst the
diamond dust episode is attributed to air masses with continental origins. This is consistent
with ARPEGE temperature and water vapour tendency favouring predominantly advection
processes including microphysical processes for water vapour.



## 1. Introduction

The impact of global warming has become obvious in high latitude regions, particularly in the Arctic region, where melting ice and softening tundra are causing profound changes. The environmental response of the Arctic is characteristically different from that of the Antarctic because of differences in planetary geography and energy circulation. Over the past 50 years, the west coast of the Antarctic Peninsula has been one of the most rapidly warming parts of the planet. This warming is not only restricted to the land but can also be noted in the Southern Ocean. For example, the warming of the Antarctic winter troposphere is more important than anywhere on Earth with a rate of 0.5 to 0.7°C per decade measured over the last thirty years (Turner et al., 2006). In Antarctica, the polar vortex is more intense, is colder and lasts longer than in Arctic. The role of the Antarctic ice is important because it is one of the key parameters in the regulation of air temperature near the surface. During the austral winter, in the absence of solar radiation, the surface cools via infrared radiation emitted towards a very cold and very dry atmosphere. In the austral summer, the absorption of solar radiation at shorter wavelength produces a diurnal cycle and warms the surface while heating is limited by a high albedo (Pirazzini, 2004; Hudson et al., 2006).

Changes in the abundance of water vapour ($H_2O$) influence directly (and indirectly via clouds) the Earth's radiation budget and therefore affect climate change (Brasseur et al., 1999) because $H_2O$ is the main greenhouse gas that emits and absorbs in the infrared domain. With an average altitude of 2500 m above sea level, the Antarctic Plateau is one of the coldest and driest places of the planet, for instance with temperature less than -80 °C and integrated water vapor amount less than 0.5 mm in winter at the Dome C station (e.g., Tomasi et al., 2012). For these reasons, numerous studies focused on climate change (e.g., Hines et al., 2004), processes in the atmospheric boundary layer (e.g., Argentini et al, 2005), reactive species





interacting with the snow (e.g., Davis et al., 2001) and astronomical site quality (e.g.,
Tremblin et al, 2011).

Clouds also play an important role in the radiation budget of the Earth. Since they have

large spatial, seasonal and diurnal variability and they are poorly represented in climate
models, large differences are obtained by climate models when assessing the strength and the
direction of the cloud feedback on the Earth radiation balance (Dufresne and Bony, 2008).
The interconnections between the Antarctic, the middle latitudes and the tropics show that
Antarctic clouds are an important part of the global climate system (Lubin et al., 1998). Based
on observations from CloudSat and Cloud-Aerosol Lidar and Infrared Pathfinder Satellite
Observation (CALIPSO) satellites over the period 2006-2010 (Adhikari et al., 2012), it is
found that the Antarctic Plateau has the lowest cloud occurrence of the Antarctic continent
(<30%). The continental region of the Antarctic Plateau experiences cloud occurrence of
about 30% at low levels (less than 3 km) and less than 10% above 5 km whilst the western
continental region records cloud occurrence of about 50% at low levels and of about 30% up
to 8 km above the surface. Furthermore, whatever the season considered, it is shown that
multilayer clouds occur over Antarctica.

The Dome C station (Concordia) in Antarctica (75°06'S, 123°21'E, 3233 m above mean

sea level) is operated jointly by the French Polar Institute Paul-Emile Victor (IPEV) and the
Italian Institute Programma Nazionale Ricerche in Antartide (PNRA). The site is located on
the Antarctic plateau with 24 hours a day in summer and 24 hours of night in winter, a
climatological temperature between -40 °C and -20 °C in summer and -80° C and -60 °C in
winter (Tomasi et al., 2006). Situated on top of a dome, there is no katabatic wind as in the
case of the costal station of Dumont d'Urville (66°S, 140°E, 0 m above sea level) since the
average wind rarely exceeds 5 m s$^{-1}$ throughout the year. When the temperature drops, water
may precipitate and light ice crystals may be suspended in the air producing a phenomenon





referred to as diamond dust. At the Dome C station, numerous studies already focused on the
diurnal and seasonal variations of the atmospheric boundary layer (e.g., Ricaud et al., 2012).

The main motivation of the present analysis is to investigate the presence of two different

clouds (thick cloud and diamond dust) that appeared above the Dome C station by combining
measurements from several instruments installed at the station, together with space-borne
measurements and model outputs. We intend to study the nature of the clouds and the
meteorological processes that favored their formation by using parameters such as
tropospheric temperature and absolute humidity, integrated water vapour, nebulosity, long-
and shortwave up- and downward radiations, together with the vertical distribution of aerosol
depolarization ratios.

We concentrate our efforts investigating two episodes: from 15 March to 8 April 2011 and

from 4 to 5 March 2013. Several instruments have been used. 1) A ground-based microwave
radiometer (HAMSTRAD, $H_2O$ Antarctica Microwave Stratospheric and Tropospheric
Radiometers) installed at Dome C that provided vertical profiles of tropospheric temperature
and absolute humidity to calculate Integrated Water Vapour (IWV) with a 7-min integration
time. 2) Daily radiosoundings launched at 12:00 UTC at Dome C. 3) A tropospheric aerosol
Lidar that provides aerosol depolarization ratio along the vertical at Dome C. 4) Down- and
upward short- and longwave radiations as obtained from secondary standard pyranometers
and pyrgeometer installed at Dome C and belonging to the Baseline Surface Radiation
Network (BSRN). 5) Space-borne aerosol depolarization ratio from the Cloud-Aerosol Lidar
with Orthogonal Polarization (CALIOP) Lidar aboard the CALIPSO platform along orbits
close to the Dome C station. The time evolution of the atmosphere over the 1-month period in
2011 has also been evaluated by considering the outputs from the meso-scale model AROME
in 3 configurations. 1) "Operational", operating mode with a snow albedo of 0.80. 2)
"Operational with ice tuning", as in "Operational" but with a setting of snow albedo that can



reach up to 0.85. And finally 3) "ARPEGE micro-physics", as in "Operational with ice
tuning" but includes the physics of ARPEGE and a state-of-the-art scheme to represent the
snow pattern taking into account the roughness length. And finally, we will use the global-
scale ARPEGE meteorological analyses in 2013 selected over the Dome C station.
The manuscript is structured as follow. Section 2 presents all the data sets used in our
study. Section 3 investigates the episode 1, namely the thick cloud and the diamond dust
episodes during the 1-month period in 2011 considering the temporal evolution of the
different parameters above and in the surroundings of the Dome C station. Section 4 deals
with the episode 2 in 2013. The genesis of the formation of the thick cloud and the diamond
dust episodes is discussed in Section 5. Finally, Section 6 concludes the study.

**2. Datasets**
**2.1. The HAMSTRAD Radiometer**
The HAMSTRAD ($H_2O$ Antarctica Microwave Stratospheric and Tropospheric
Radiometers) instrument is a state-of-the-art microwave radiometer to probe the troposphere
in very cold and very dry environments in order to retrieve temperature and absolute humidity
vertical profiles, and IWV. Temperature profiles are obtained from the 51–59 GHz spectral
range, centered on the oxygen line. Absolute humidity profiles are retrieved from the 169–197
GHz spectral range, centred on the water vapour line. IWV is calculated from the water
vapour profile integrated along the vertical. Integration time is 7 min. The radiometer is
presented in Ricaud et al. (2010).
The instrument was sent to Dome C in January 2009. It has been running automatically
since January 2010. Science and validation studies using HAMSTRAD data are detailed in
Ricaud et al., 2012; 2013; 2014a-c; and 2015. All the HAMSTRAD data measured since 2009
are      freely      available      at      the      following      address:



http://www.cnrm.meteo.fr/spip.php?article961&lang=en. The radiometer sensitivity is very
high in the planetary boundary layer, high in the free troposphere and very weak in the upper
troposphere-lower stratosphere (Ricaud et al., 2015). The $H_2O$ and temperature vertical
resolutions are ~20-50 m, ~100 m and ~500 m in the planetary boundary layer, in the free
troposphere and in the upper troposphere-lower stratosphere, respectively. Against
radiosondes, there is a 1-5 K cold bias below 4 km, and a 5-10 K warm bias above. There is a
wet bias of 0.1-0.3 g m$^{-3}$ below about 2 km and a dry bias of ~0.1 g m$^{-3}$ above. Integrated
water vapour is of a high quality, 1-2% wetter than radiosondes.

**2.2. Radiosondes**
The programme of radiosoundings developed at Dome C is presented in Ricaud et al.
(2014a). Temperature and humidity biases against HAMSTRAD are shown in the previous
section. In the present study, the vertical profiles of temperature and humidity were taken
from RS92 radiosondes using the standard Vaisala evaluation routines without any correction
of sensor heating or time lag effect. We recall that the corrections performed on the
radiosonde data measured in 2009 according to Miloshevish et al. (2006) shown a weak
impact (with a maximum of +4% on IWV) on the vertical profiles (Ricaud et al., 2013).
Furthermore, considering the updated tools developed in Miloshevich et al. (2009), Tomasi et
al. (2011 and 2012) found that, between 630 and 470 hPa, the correction factor for humidity
estimated by the radiosonde varied within 1.10-1.15 for daytime and within 0.98-1.00 for
nighttime. It is important to note that the 630-470 hPa layer is located between the ground and
an altitude of ~2 km which maximizes the calculation of IWV. A 1.2 K cold bias is also
observed in the RS92 from the surface up to an altitude of ~4 km (Tomasi et al., 2011 and

2012).






**2.3. The Aerosol Lidar**
The aerosol lidar is a backscatter and depolarization system in operation at Dome C in
relation with different scientific projects (http://lidarmax.altervista.org/lidar/ Antarctic
LIDAR.php). Vertical profiles of aerosol and cloud structures are continuously measured
together with the characterization of the physical phase of particles. Automated daily images
are produced and sent to Italy to monitor the state of the atmosphere above Dome C and to
check the instrument operations.
The Lidar system uses a Laser Quantel (Brio) at Dome C and operates at 532 nm to get
backscattering and depolarization ratio from 30 to 7000 m above ground with a 7.5-m vertical
resolution. The line of sight is zenith looking through a window enabling measurements in
all-weather conditions. The telescope has a 10-cm diameter, with 30-cm refractive optics and
0.15-nm interference filter. It has already been used in several scientific studies, e.g. the ra-
diative properties of $H_2O$ and clouds in the far infrared over Antarctica (Palchetti et al., 2015).

**2.4. CALIOP onboard CALIPSO**
The CALIPSO satellite has been launched to study the role clouds and aerosols play in the
Earth system that includes air quality, weather and climate. CALIPSO was launched on 28
April 2006 with the cloud profiling radar system on the CloudSat satellite. The CALIPSO
satellite comprises three instruments, the CALIOP Lidar, the Imaging Infrared Radiometer
(IIR), and the Wide Field Camera (WFC) (Winker et al., 2009).
CALIOP is a two-wavelength (532 nm and 1064 nm) polarization-sensitive lidar that
provides high-resolution vertical profiles of aerosols and clouds
(https://calipso.cnes.fr/en/CALIPSO/lidar.htm). CALIOP uses three receiver channels: one
measures the 1064 nm backscatter intensity and two channels measure orthogonally polarized
components of the 532 nm backscattered signal. The receiver telescope is 1 metre in diameter.





The full-angle field of view of the telescope is 130 µrad resulting in a footprint at the Earth's
surface of about 90 metres. Algorithms have been developed to retrieve aerosols and cloud
layers together with optical and microphysical properties (Young and Vaughan, 2009).
Depolarization ratio estimated with version V3.01 is presented in our study.

**2.5. The BSRN Network**

The objective of the World Climate Research Programme (WCRP) Baseline Surface

Radiation Network (BSRN) is to provide, using a high sampling rate, observations of the best
possible quality, for short- and longwave surface radiation fluxes. These readings are taken
from a small number of selected stations, including Dome C, in contrasting climatic zones,
together with collocated surface and upper air meteorological data and other supporting
observations. The incoming longwave and shortwave radiation components of the surface
radiative balance were taken from the Dome C BSRN station, and measured with two Kipp &
Zonen CM22 secondary standard pyranometers and two Kipp & Zonen CG4 Pyrgeometers,
all operated according to BSRN guidelines (Lanconelli et al., 2011).

**2.6. The AROME Model**

AROME (Seity et al., 2011) is a small-scale numerical prediction model, operational at

Meteo-France since December 2008. It was designed to improve short-range forecasts of
severe events such as intense Mediterranean precipitations (Cévenole events), severe storms,
fog, urban heat during heat waves. The physical parameterizations of the model come mostly
from the MESO-NH model whereas the dynamic core is the ALADIN model. The size of the
mesh is 2.5 km against 10 km for ARPEGE over France in 2014. The model is initialized
from a 3D-var data assimilation system using radar reflectivity and Doppler wind. Five daily
forecasts are made with AROME, thus helping to better predict meteorological events of the





day and of the morrow (30 h forecast range). AROME was used within the project GEWEX
Atmospheric Boundary Layer Study 4 (GABLS4) whose one of the motivations was the study
of the meteorological evolution over the Dome C station (Bosveld et al., 2014).

For the study, three experiments at 2.5 km were used. Two based on the AROME con-

figurations: 1) "Operational", operating mode with the default snow scheme (Douville et al.,
1995)  (labeled as 79HA). 2) "Operational with ice tuning", as in "Operational" but with a set-
ting of a minimum snow albedo of 0.8 (labeled as 79YG). In the third one "ARPEGE micro-
physics", the AROME physics was replaced by the ARPEGE one used in the global model
with the state-of-the-art scheme to represent the permanent snow with a minimum snow
albedo of 0.8 valid over Dome C and a more accurate roughness length (labeled as 79Z6).

**2.7. The ARPEGE Model**

The ARPEGE model is the global model used for the numerical weather prediction

(NWP) at Météo-France. In the present study, the operational configuration has been used
with a stretched grid at high resolution over France (10 km) and a coarser grid over Australia
of 60 km. At the South Pole, the horizontal resolution is about 50 km and the vertical grid has
70 levels with a first level at around 16 m above the ground. The assimilation tool is based on
an incremental 4-dimensional variational (4D-Var) method. The physical package used in the
ARPEGE model is at the state-of-the-art, with a Turbulent Kinetic Energy scheme associated
with a mass flux scheme for the boundary layer (Bazile et al., 2011). The clouds and the
micro-physics use 4 prognostic variables such as cloud water, cloud ice, rain and snow
(Lopez, 2002; Bouteloup et al., 2005). The radiative transfer in the atmosphere is computed
with the Rapid Radiative Transfer Model (RRTM) scheme (Mlawer et al., 1997) for the
longwave and the shortwave with the Fouquart-Morcrette scheme (Fouquart and Bonnel,
1980; Morcrette et al., 2001).




## 3. Episode 1 on 15 March-8 April 2011

### 3.1. Temperature

Figure 1 shows the time evolution of temperature from 15 March to 8 April 2011 in the
planetary boundary layer at 4, 50 and 100 m above the ground as measured by HAMSTRAD,
the radiosondes and as calculated by AROME according to the 3 configurations detailed in
the previous section, namely a) operational, b) operational with ice tuning and c) considering
ARPEGE micro-physics. Unless explicitly specified, from now, all the heights refer to height
above the ground. The time evolution of temperature profiles as measured by HAMSTRAD
from 0 to 5 km is shown Fig. 2.
In the planetary boundary layer (Fig. 1), temperatures from HAMSTRAD and the
radiosondes are rather stable from 15 to 24 March 2011 (210-220 K at 4 m), rapidly increase
to ~240 K on 25 March and start decreasing on 26 March to 31 March (~210 K). After a
slight increase on 4 April (~220 K), a minimum in temperature is reached on 5 April (210 K
at 50 and 100 m) with stable temperature at 4 m while increasing at 50 and 100 m reinforcing
the inversion. The sharp increase in temperature measured at 4 m (~30 K) on 25 March is
much less intense above, at 50 and 100 m (~5 K), but the sharp decrease in temperature on 5
April is much more intense at 50 and 100 m (~20 K) than at 4 m (~10 K maximum). At 50
and 100 m, measurements from HAMSTRAD and radiosondes are very consistent with the
outputs from AROME whatever the configuration considered. But there are some differences
at 4 m. The biggest differences are detected during the period 20-25 March, prior to the warm
period 25-26 March, during which AROME outputs are systematically greater than the
measurements by 10-15 K whatever the configuration considered. During the other periods,
although the operational AROME outputs (79HA) are warmer than the outputs from the two
other configurations by 2-3 K on average at 06:00 UTC (14:00 LT), the outputs from





AROME version ARPEGE micro-physics (79Z6) around 16:00 UTC (local midnight) is
warmer than the two other configurations on few periods (16-25 March and 2, 4, 7 and 9
April). At 12:00 UTC (20:00 LT), the outputs from 79Z6 are the closest to the radiosonde and
the HAMSTRAD measurements. We note a diurnal cycle in temperature of ±3 K observed in
the AROME outputs from the operational configuration that is not present in the
HAMSTRAD data set. Above, at 50 and 100 m, all the data sets considered are very
consistent to each other within ±2 K.

From 0 to 5 km (Fig. 2), the two episodes of abrupt changes in temperature are clearly

detected on 25 March and on 5 April 2011, with substantial increase and decrease in
temperature, respectively. Along the vertical, these two episodes cover a wide domain from
the ground to more than 3 km. The warm episode lasts 3-4 days although the cold episode is
of a short duration, namely 1 day.

**3.2. Integrated Water Vapour**

If we now consider the evolution of IWV over the same period (Fig. 3), we note a slight

positive change from 15 to 22 March 2011 (from 0.3 to 0.6 kg m$^{-2}$), followed by an abrupt
increase on 25 March of 1.0 kg m$^{-2}$ in less than 24 hours. After a 2-day plateau at 1.4 kg m$^{-2}$
in HAMSTRAD and radiosonde data, IWV decreases slowly until the end of the period.
Neverthless, on 5 April, the atmosphere is the driest of the period reaching 0.1 kg m$^{-2}$ within
few hours. All the data sets (HAMSTRAD, radiosonde and AROME) behave consistently
during this period. HAMSTRAD and the radiosonde data do not exhibit any bias, whilst
AROME outputs tend to show a much wetter atmosphere compared to HAMSTRAD and the
radiosondes of about 0.2 kg m$^{-2}$, except during the warm and wet period (25-26 March) when
the bias is even greater reaching 0.4-0.5 kg m$^{-2}$.





### 3.3. Absolute Humidity


Figure 4 shows the time evolution of absolute humidity vertical profiles from 15 March to
8 April 2011 in the planetary boundary layer (4, 50 and 100 m) as measured by HAMSTRAD,
the radiosondes and as calculated by AROME according to the 3 configurations. The time
evolution of absolute humidity as measured by HAMSTRAD from the ground to 5 km is
shown Fig. 5.
Consistently with the time evolution of IWV, the evolution of absolute humidity in the
planetary boundary layer (Fig. 4) and in the free troposphere (Fig. 5) shows an abrupt increase
on 25 March and a net decrease on 5 April, but with lots of differences within all the datasets
and the altitude layers considered. Prior to the warm and wet episode of 25-26 March,
HAMSTRAD measurements from 4 to 100 m (Fig. 5) are systematically much wetter than
both the radiosondes and the AROME outputs by 0.1-0.2 g m$^{-3}$ and 0.4-0.5 g m$^{-3}$,
respectively. It is a very well documented bias already presented in several works (e.g. Ricaud
et al., 2014a and 2015) that HAMSTRAD measures a wetter atmosphere below 2 km and
drier above, than any other data sets (radiosonde, in situ, space-borne, and meteorological
analyses). After the warm and wet period, the bias reduces to 0.1 g m$^{-3}$ between HAMSTRAD
and the radiosondes. We note some differences in the $H_2O$ time evolution within the AROME
outputs according to the 3 configurations. The operational outputs (79HA) are usually much
wetter than the two other configurations, particularly around local noon and in the lowermost
layer at 4 m. The biases within the outputs of the 3 configurations decrease with height.
Along the vertical (Fig. 5), the time evolution of the $H_2O$ field obviously shows the two
episodes of abrupt changes detected on 25 March and on 5 April 2011, with a net increase and
a net decrease in $H_2O$, respectively. These two episodes cover a wide domain from the ground
up to 2.5-3 km. Consistently with the conclusions drawn with temperature evolution in the



section 3.1, the wet and warm episode (see Fig. 2) lasts 2-3 days although the cold and dry
episode is of a short duration, namely less than 1 day.

**3.4. Radiation**
The time evolution of the downward and upward short- and longwave radiations as
measured by the BSRN international network is displayed in Figure 6, together with the net
irradiance (difference between the downward and the upward fluxes) from 15 March to 8
April 2011. The diurnal cycle of solar irradiance fluxes is clearly evidenced with the obvious
maximum at local noon ranging between 350 W m$^{-2}$ (at the beginning of the period) and 150
W m$^{-2}$ (at the end of the period). Albedo over the whole period is found to range between 0.8
and 0.95 with daily minimum at local noon (not shown). The upward longwave radiation
emitted by the surface is generally greater than the downward irradiance in clear sky
conditions, while they became similar under overcast when thick cloudiness prevents
radiative cooling. Consequently, alternating day and night periods in March and April, the net
irradiance is negative except around local noon when it can be either positive or close to zero.
But for the two periods considered so far, namely on 25-26 March (warm and wet) and on 5
April (cold and dry), the radiation budget is significantly different from the average situation.
On 25 March, the longwave radiations (both downward and upward) are much greater
than the shortwave radiations (both downward and upward) even at local noon. The resulting
effect is that the net irradiance is positive or close to zero over the whole period. This
obviously indicates that a thick cloud is shielding the downward shortwave radiation (coming
from the Sun) and increases the downward longwave radiation (coming from the cloud).
Furthermore, there is a great probability a thick cloud is present over the Dome C station
during the warm and wet period.





On 5 April during the cold and dry period, the situation is radically different. There is not
an abrupt increase of longwave downward radiation as it was the case during the warm and
wet period, so we can rule out the presence of a thick cloud above the station. Nevertheless,
the situation is atypical since, at local noon, the downward shortwave radiation is only slightly
greater than the upward shortwave radiation, and the net irradiance does not exhibit an
obvious diurnal cycle maximizing at local noon. Consequently, even if the presence of a thick
cloud has been ruled out from the longwave radiation analysis, both the shortwave and the
total irradiance analyses tend to suggest the presence of a cloud, probably thin and/or close to
the surface, in order to 1) slightly affect the downward longwave irradiance, and 2) strongly
affect the diurnal cycle of the net irradiance. The next section investigates the presence and
the nature of the clouds during the two periods under consideration: a) the warm and wet
period (25-26 March 2011) and b) the cold and dry period (5 April 2011).

**3.5. Clouds**
The time evolution of the nebulosity vs. height as calculated by AROME according to the
3 configurations over the period 25 March-8 April 2011 is displayed Figure 7. In two
configurations (operational and operational with ice tuning), clouds (traced by values of
nebulosity greater than 0.25) are calculated mainly over two single periods: 1) on 22 March
from 1 to 5 km, and 2) on 25-29 March from the ground to 6 km. Considering the third
configuration (ARPEGE microphysics), the period when clouds (traced by values of
nebulosity greater than 0.25) are present is much longer than the two first configurations since
it almost covers the entire time interval under consideration. There are indeed the two periods
previously cited, namely on 22 March, and 25-29 March, but they extend both in time (22-24
March and 25-30 March, respectively) and altitude (surface to 7 km). Other periods show
some moderate values of nebulosity (0.10-0.25) on 15-17, 20 March, and 2, 3, 4, 6-7 April.



These calculated clouds can be close to the ground or high in the free troposphere (4-6 km). In
the three configurations, high nebulosity values (greater than 0.8) are calculated close to the
surface (0-200 m). The AROME model tends to produce a sort of cloud residual in the
planetary boundary layer in the three configurations analyzed.
The warm and wet period (25-26 March 2011) highlighted in the previous sections is
indeed characterized by high values of nebulosity (greater than 0.8) whatever the
configurations of AROME considered, from the surface to 6-7 km. This is another indicator
of the presence of a thick cloud over the Dome C station during that period. The cold and dry
period (5 April 2011) is nevertheless not characterized by such a high value of nebulosity
extending in the free troposphere but rather by some high nebulosity being confined below
100 m. But this is probably an artifact of all the runs performed by ARPEGE within the 3
configurations, producing a residual cloud in the lowermost planetary boundary layer.
In order to check whether clouds are present or not over the station during the two periods
studied in detail, we consider now the time evolution of the aerosol depolarization as
measured by the Lidar installed at Dome C on 24-26 March and on 4-6 April 2011 (Fig. 8).
For the warm and wet period (Fig. 8 top), high depolarization ratios (greater than 30%),
signature of ice particles, start increasing by the end of 24 March (22:00 UTC), reaching an
altitude of 1.2-1.5 km, all over 25 March, and start decreasing on 26 March by 12:00 UTC.
The vertical structures in the depolarization ratio fields are a signature of precipitation of ice
particles (Mishchenko et al., 2000). On 26 March, at about 2 km altitude from 07:00 to 13:00
UTC, a layer of low depolarization ratio (less than 14%) appears, that is a signature of liquid
water cloud. In general, over this period, the cloud is so opaque that the Lidar signal cannot
penetrate the structure beyond ~1.2 km altitude.
We may have a better insight in the vertical structure of the cloud covering the Dome C
station on that warm and wet period by considering the space-borne CALIOP Lidar nighttime





measurements performed on 25 March 2011 in the vicinity of the station. Figure 9 top left
shows the Spaceborne Lidar CALIOP measurements of depolarization ratio along one orbit in
the vicinity of the Dome C station on 25 March 2011. We note that, at the location of the
Dome C station (75°06'S, 123°21'E), the depolarization ratio is greater than 0.4 (ice particles)
from the ground (3233 m above mean sea level, amsl) to about 10 km amsl, namely ~7 km
above the ground. If we now combine the downward and upward Lidar information, we can
state that on the warm and wet period (24-26 March 2011), a 7-km thick ice cloud passed over
the Dome C station and precipitated ice particles whilst, by the end of the period, a low-
altitude (~2 km) liquid water cloud was also present.
For the cold and dry period (centered on 5 April 2011), high depolarization ratios (greater
than 30%) from the Lidar operating at Dome C (Fig. 8 bottom) start increasing by the middle
of 4 April (12:00 UTC), reaching a maximum altitude of 100 m, increasing up to 200 m on 5
April at 09:00 UTC, to finish decreasing on 6 April by 12:00 UTC. The high depolarization
ratio suggests that the cloud is constituted of ice crystals and, since there is no vertical layers
(as during the warm and wet period), there is no trace of precipitation. Considering the
CALIOP space-borne Lidar measurements of depolarization ratio (Fig. 9 top right) performed
on 5 April 2011 in the vicinity of the Dome C station, they also suggest a much thinner cloud
from the ground to about 4 km amsl, namely less than 1 km above the ground, with values
ranging 0.1-0.2 (ice particles). Since there is no precipitation and no presence of standard
thick clouds above the station, the thin cloud episode is traditionally attributed to a diamond
dust episode, rather frequent at the Dome C station.
Diamond dust is usually made of ice crystals in suspension in the air located in the
lowermost troposphere. At the South Pole station, when sorted by number, Lawson et al.
(2006) attributed 45% of the ice crystals recorded to diamond dust (columns, thick plates and
plates), 30% are rosette shaped (mixed-habit rosettes, plate-like polycrystals and rosette





shapes with side planes) whilst 25% are irregular. In the Eastern Antarctic Plateau over all the
seasons except summer, a strong surface-based temperature inversion persists in which
vertical mixing causes the boundary-layer air to become supersaturated with respect to ice.
Consequently, small ice crystals referred to as diamond dust form in this layer (Walden et al.,
2003). Nevertheless, even in the absence of mixing, longwave cooling of the near-surface air
can also lead to supersaturation with respect to ice and form ice crystals. Classification of ice-
cloud particles is important to retrieve the shape of individual crystals and to estimate the
radiative impact of the clouds (Bailey and Hallett, 2009; Lindqvist et al., 2012). It is beyond
the scope of the present analysis to classify ice crystals measured over the Dome C station.

## 4. Episode 2 on 4-5 March 2013

The second episode is much shorter than the first one since it lasts only two days from 4 to
5 March 2013. It relies on the same datasets as presented in section 3 except that the analyses
are from the meteorological operational model ARPEGE that routinely delivers since
December 2011 every 6 hours (00:00, 06:00, 12:00, and 18:00 UTC) meteorological fields at
the vicinity of the Dome C station.

### 4.1. Temperature

The temperature anomaly over the 2-day period is represented in the Figure 10 as
measured by HAMSTRAD and as calculated by ARPEGE. From the surface to about 2 km
altitude, they both show a warm period on 4 March followed by a cold period on 5 March,
with a transition propagating in the HAMSTRAD data up to 4 km altitude, probably due to
the vertical resolution of the microwave radiometer measurements. Above this altitude, a cold
period is followed by a warm period in the two datasets. Although the HAMSTRAD data are
noisier than the ARPEGE data, the maxima and minima are consistently observed and



447 calculated in the lowermost troposphere around 12:00-18:00 UTC on 4 March and around

448 18:00-22:00 UTC on 5 March, respectively.


450 **4.2. Water Vapour**

451  The water vapour anomaly over the 2-day period is represented in the Figure 11 as

452 measured by HAMSTRAD and as calculated by ARPEGE. From the surface to about 4-5 km

453 altitude, they both show a wet period on 4 March followed by a dry period on 5 March. As for

454 temperature, the HAMSTRAD $H_2O$ data are noisier than the ARPEGE data, but the maxima

455 and minima are consistently observed and calculated in the lowermost troposphere around

456 12:00-18:00 UTC on 4 March and around 18:00-22:00 UTC on 5 March, respectively.

458 **4.3. Radiation**

459  The time evolution of the downward and upward short- and longwave radiations as

460 measured by the BSRN international network is displayed in Figure 12, together with the net

461 irradiance (difference between the downward and the upward fluxes) from 1 to 9 March 2013.

462 As already presented in section 3.4, the diurnal cycle of solar irradiance fluxes mainly shows

463 a clear-sky period over the Dome C station, except over the period from 4 March at 00:00

464 UTC to 6 March at 00:00 UTC. Indeed, on 4 March all day long, there is a net increase in the

465 longwave downward radiation from 80 to 120 W m$^{-2}$ compared to the values from 1 to 3

466 March and from 7 to 9 March when we can expect the station is under clear sky conditions.

467 Furthermore, from 12:00 to 24:00 UTC, the net irradiance is about –20 W m$^{-2}$ on 4 March,

468 whilst it is usually about –30 W m$^{-2}$ in clear sky conditions. Consequently, there is a great

469 probability a thick cloud is present over the Dome C station on 4 March during 24 h.

470  On 5 March, between 12:00 and 24:00 UTC, the net irradiance is very low (about –50 W

471 m$^{-2}$) compared to values of about –30 W m$^{-2}$ in clear sky conditions. There is a slight increase



of the longwave downward (90 W m$^{-2}$) and upward (140 W m$^{-2}$) fluxes on 5 March compared
to the fluxes in clear sky conditions (70 and 120 W m$^{-2}$, respectively), but much less than
fluxes in cloudy conditions (150 and 160 W m$^{-2}$, respectively). Consequently, this period of
12 hours on 5 March cannot be attributed to neither clear sky nor thick cloud episodes.

**4.4. Clouds**
Now we consider the presence of clouds and/or ice/liquid particles over the 2-day period
either from active and passive remote-sensing measurements or from ARPEGE analyses. The
time evolution of ice water mixing ratio calculated by ARPEGE over Dome C is represented
in the Figure 13 top together with the total precipitation flux over the 2-day period (Fig. 13
bottom). ARPEGE analyses obviously calculate ice cloud from the surface to an altitude of
about 4 km on 4 March, with a top altitude decreasing down to the surface on 5 March at
12:00 UTC. Between 18:00 and 24:00 UTC on 5 March, there is also a trace of ice cloud from
0 to 1 km altitude. The main thick cloud calculated on 4 March is associated with ice
precipitation from the altitude of ~3 km at 06:00 UTC down to ~2 km on 20:00 UTC (Fig. 13
bottom). There is no longer trace of local precipitation after 03:00 UTC on 5 March.
The depolarization ratio measured by the aerosol Lidar installed at Dome C from 4 to 5
March is shown on Figure 14. For the warm and wet period (4 March), high depolarization
ratios (greater than 30%) are present all over 4 March, reaching an altitude of 1.5-2.0 km, and
start decreasing on 5 March by 00:00 UTC. The vertical structures in the depolarization ratio
fields are a signature of precipitation of ice particles. Between 13:00 and 14:00 UTC on 4
March and from 00:00 to 10:00 UTC on 5 March, depolarization ratios are much lower,
reaching values of less than 10%. This is clearly the signature of the presence of supercooled
liquid water. From 10:00 to 24:00 UTC on 5 March, depolarization ratios are very high
(>40%), indicative of ice crystals, but confined from the surface to 100-200 m altitude. There



is no vertical structures, it means there is no precipitation associated to the presence of the
cloud. Furthermore, this ice crystals can be considered as in suspension in the air and labeled
as "diamond dust". This is confirmed by the BSRN radiation measurements (see section 4.3).
Over this 2-day period, only one CALIOP/CALIPSO orbit has been analyzed in the
vicinity of the Dome C station (Fig. 9 bottom) on 4 March (08:15 UTC) during the thick-
cloud episode. On that day, the depolarization ratio is ranging 0.1-0.3 from the ground (3233
m amsl) to about 8 km amsl, namely ~5 km above the ground. Note that there is no CALIPSO
orbit in the vicinity of the Dome C station in coincidence with the diamond dust episode.
If we synthesize our findings relative to the episodes 1 and 2, we can state the following.
The time evolution of temperature, absolute humidity, ice and aerosol fields obviously shows
two episodes of abrupt changes. Firstly, a warm and wet period is associated with a thick
cloud that develops from the surface to 5-8 km and is constituted of ice crystals that
precipitate. Secondly, a cold and dry period is associated with a thin cloud that develops close
to the surface (100-200 m) and is constituted of ice crystals in suspension in the air. This later
episode is known as "diamond dust" episode.

**5. Discussions**
In this section, we investigate the processes that contributed to the presence of a thick-
cloud and a diamond-dust episode above the Dome C station considering the origin of air
masses, the integrated water vapour fields over Antarctica and the temperature and water
vapour budgets calculated by ARPEGE.

**5.1. Origin of Air Masses**
The impact of the origin of air masses on the short-term variability of $H_2O$ and temperature
and the high correlation coefficient (greater than 0.90) between water vapour and temperature



at Dome C over the entire year 2010 were presented in Ricaud et al. (2012 and 2014c) based
on 5-day back-trajectory calculations. We propose, in the present study, to use the same
methodology to interpret the time evolution of the atmosphere during the two above-
mentioned episodes. We have thus considered a 5-day back trajectory study based upon the
European Centre for Medium-Range Weather Forecasts (ECMWF) analyses starting from the
Dome C location at five different pressure levels from the planetary boundary layer (650 and
600 hPa), to the free troposphere (500, 400 and 300 hPa).
For episode 1, Figure 15 (top left) shows the 5-day back-trajectories at the 5 selected
pressure levels during the warm and wet period (see the section 3) on 25 March 2011 at 12:00
UTC corresponding to the time of maximum temperature and absolute humidity of an air
parcel issued from Dome C. In the lowermost layers (650 and 600 hPa), the air parcels are
mainly issued from the Antarctic continent. But higher up, at 500, 400 and 300 hPa, air
masses are coming from the oceanic middle latitudes, between Australia and New Zealand,
imprint of warm and wet air masses. When air masses in the free troposphere reach the
Antarctic continent, they are uplifted and temperature decreases by more than 50 K (not
shown). Note the air parcel at 400 hPa that is firstly originated from oceanic high latitudes in
the vicinity of the Antarctic continent but moves towards the middle oceanic latitudes with a
net subsidence and an increase of temperature of 30 K.
Regarding the diamond-dust period, Figure 15 (top right) shows the 5-day back-trajectories
at the 5 selected pressure levels on 5 April 2011 at 12:00 UTC corresponding to the time of
minimum temperature and absolute humidity of an air parcel issued from Dome C. We can
note that all the calculated air masses are originated from the Antarctic plateau whatever the
pressure level considered. Consequently, as already studied in Ricaud et al. (2012 and 2014c),
we thus expect that both temperature and $H_2O$ tends to decrease on 5 April at 12:00 UTC



compared to the surrounding periods because air masses with continental origins produce a
cold and dry atmosphere above Dome C (as on 5 April 2011).
For episode 2, Figure 15 bottom left shows the 5-day back-trajectories at the 4 selected
pressure levels of 650, 600, 500 and 400 hPa during the warm and wet period (see section 4)
on 4 March 2013 at 08:00 UTC corresponding to the time of maximum temperature and
absolute humidity of an air parcel issued from Dome C. At 650 hPa, the air parcel has a
continental origin but migrates very close to the coast 2 days before reaching Dome C.
Above, at 600, 500 and 400 hPa, all the air masses are coming from the oceanic middle-high
latitudes ranging from 47°S to 63°S and from the surface to ~680 hPa, namely imprint of
warm and wet air masses. As for episode 1, when air parcels in the free troposphere reach the
Antarctic continent, they are uplifted and temperature decreases by 20-30 K (not shown). On
5 March at 18:00 UTC (Fig. 15 bottom right) during the cold and dry period of episode 2
corresponding to the time of minimum temperature and absolute humidity, the meteorological
situation is radically different. Whatever the pressure level considered, the air parcels are all
confined to the Antarctic plateau in the vicinity of the Dome C station, explaining again, as
for episode 1, the cold and dry atmosphere observed during episode 2.

**5.2. Integrated Water Vapour over Antarctica**
If we consider the IWV fields as calculated by the NCEP/NCAR operational analyses
(Kalnay et al., 1996) on 25 March 2011 and 5 April 2011 for the episode 1 over the Antarctic
continent (Figure 16 top left and right, respectively), we obviously remark that the Eastern
Antarctic plateau is much wetter on 25 March than on 5 April. The IWV calculated over the
Dome C station is ~1.4 kg m$^{-2}$ on 25 March 2011 and ~0.6 kg m$^{-2}$ on 5 April 2011 in excellent
agreement with the HAMSTRAD measurements (see section 3.2). This is indeed induced by
the oceanic-origin flux bringing warm and wet air masses over the Dome C station on 25



March and by the continent-origin flux bringing cold and dry air masses over the station on 5
April. The same exercise can be performed for the episode 2 (Figure 16 bottom left and right,
respectively) considering the IWV fields as calculated by the NCEP/NCAR operational
analyses on 4 and 5 March 2013. There, the IWV fields above Dome C show a similar pattern
between 4 and 5 March but with a slight wet inflection on 4 March compared to 5 March with
daily averaged values of ~0.6 and ~0.4 kg m$^{-2}$, respectively consistent with the daily-averaged
values obtained at Dome C with HAMSTRAD (~0.55 and ~0.30 kg m$^{-2}$, respectively).
Consequently, considering episodes 1 and 2, the thick-cloud episode observed during the
warm and wet period above Dome C is attributed to air masses with an oceanic origin whilst
the diamond dust episode occurring during the cold and dry period is attributed to air masses
with continental origins.

### 5.3. Temperature and Water Vapour Budgets

We now intend to assess the tendency of temperature calculated by ARPEGE during
episode 2 into radiation, turbulence, microphysics, and total advection and the tendency of
water vapour into turbulence, microphysics, and total advection. Figure 17 shows the
temperature budget calculated on 4 March 2013 over the warm and wet 12-h period 00:00-
12:00 UTC and on 5 March 2013 over the dry and cold 12-h period 06:00-18:00 UTC whilst
Figure 18 focuses on the water vapour budget.
For altitudes greater than ~100 m (3333 m amsl) above the ground on 4 March and greater
than ~200 m on 5 March, the temperature tendency of the warm (Fig. 17 left) and of the cold
(Fig. 17 right) periods is mainly dominated by the advection processes (red lines). This is
fully consistent with the interpretation of the origin of air masses (previous subsection).
Nevertheless, in the planetary boundary layer below approximately 100 m on 4 March and
below 200 m on 5 March, the temperature tendency of the two periods is also governed by the



vertical mixing done by the turbulent processes (green line). Indeed, turbulence always tends
to stabilize the atmosphere impacted by radiative or dynamical forcing. The effect of the
radiative cooling on the surface temperature and its impact on the boundary layer is clearly
shown on 5 March (Fig. 17 right).

As for temperature, the water vapour tendency of the two periods also needs to be

separated at ~100 and ~200 m above the ground on 4 and 5 March, respectively. Above these
two limits, the water vapour tendency of the warm period (Fig. 18 left) and of the cold (Fig.
18 right) periods is governed by both the advection and the microphysical processes. On 4
March, a warmer and more humid air is advected (total water vapour tendency on Fig. 18 left
and total temperature advection on Fig. 17 left are positive), so the microphysics tend to
create some clouds by condensation (negative microphysics tendency, blue line on Fig. 18
left) with small precipitations close to the surface (~18 mm in 12 hours, not shown). On 5
March (Fig. 18 right), the water vapour advection (red line) is negative so a drier air is
advected toward the Dome C station. Below ~200 m, advection, turbulence and microphysical
(precipitation) processes compete to dehydrate the planetary boundary layer.

In general, this reinforces our conclusions of thick cloud episodes driven by warm and wet

air masses of oceanic origin and of diamond dust episodes driven by cold and dry air masses
of continental origin. Nevertheless, in the planetary boundary layer below approximately
~100-200 m, the water vapour tendency of the two periods is competing between advection,
microphysical and turbulence processes.

## 6. Conclusions


The present study takes the opportunity of combining several measurements and model

outputs to study the short-term evolution of the Antarctic atmosphere above the Dome C
station focusing on episodes of thick cloud and diamond dust. From 15 March to 8 April 2011



and from 4 to 5 March 2013, the atmosphere has been probed by several instruments. 1) A
ground-based microwave radiometer (HAMSTRAD) installed at Dome C that provided
vertical profiles of tropospheric temperature and absolute humidity to calculate Integrated
Water Vapour (IWV) with a 7-min integration time. 2) Daily radiosoundings launched at
12:00 UTC at Dome C. 3) A tropospheric aerosol Lidar that provides aerosol depolarization
ratio along the vertical at Dome C. 4) Down- and upward short- and longwave radiations from
an instrument installed at Dome C belonging to the BSRN network. 5) Space-borne aerosol
depolarization ratio from the CALIOP Lidar aboard the CALIPSO platform along orbits close
to the Dome C station.
The time evolution of the atmosphere over the 1-month period in 2011 has also been
evaluated by considering the outputs from the mesoscale model AROME in 3 configurations.
1) "Operational", operating mode with a snow albedo of 0.80. 2) "Operational with ice
tuning", as in "Operational" but with a setting of snow albedo that can reach up to 0.85. And
finally 3) "ARPEGE micro-physics", as in "Operational with ice tuning" but includes the
physics of ARPEGE and a state-of-the-art scheme to represent the snow pattern taking into
account the roughness length. The ARPEGE global-scale meteorological model analyses gave
the state of the atmosphere and relevant prognostics (ice precipitation, temperature and water
vapour budget) on 4 and 5 March 2013.
Two distinct periods are highlighted by all the datasets: the warm and wet periods (24-26
March 2011 and 4 March 2013) and the cold and dry periods (5 April 2011 and 5 March
2013). Although the time evolution of temperature in the planetary boundary layer and in the
free troposphere is consistent within all the data sets, AROME in 2011 tends to model a
warmer atmosphere during these two specific events at 4 m. The time evolution of absolute
humidity is also consistent within all the data sets with some known wet bias in HAMSTRAD
compared to radiosondes in the planetary boundary layer, and with some systematic wet bias



of AROME compare to radiosondes. In general, IWV from HAMSTRAD and radiosondes are
consistent with each other although AROME tends to be much wetter than the two
measurements. ARPEGE analyses in 2013, consistently with HAMSTRAD data, reproduce
the warm and wet period, and the cold and dry period.

Since the longwave radiations (both downward and upward) are much greater than the

shortwave radiations (both downward and upward) during the warm and wet period of 2011,
the effect is that the net irradiance is positive or close to zero. This obviously indicates that a
thick cloud is shielding the downward shortwave radiation (coming from the Sun) and
increases the downward longwave radiation (coming from the cloud). During the cold and dry
periods, there is not an abrupt increase of longwave downward radiation but the downward
shortwave radiation is only slightly greater than the upward shortwave radiation, and the net
irradiance does not exhibit an obvious diurnal cycle maximizing at local noon. Consequently,
both the shortwave and the total irradiance analyses tend to suggest the presence of a cloud.

Considering upward and downward active measurements of aerosols from two Lidars

installed at Dome C and aboard a spaceborne platform, respectively, the signature of a thick
cloud with high depolarization ratios (greater than 30%) is detected during the warm and wet
periods from the surface to ~5-7 km with precipitation of ice particles and the presence of a
supercooled liquid water cloud with low depolarization ratios (~10%). During the cold and
dry periods, high depolarization ratios (greater than 30%) to a maximum height of 100-500 m
is measured suggesting that the cloud is constituted of ice crystals with no trace of
precipitation. This means ice crystals are in suspension in the air. This case is usually referred
to as "diamond dust".

The presence of a thick cloud during the warm and wet period of 2011 is calculated by the

2D nebulosity fields from AROME extending from the ground to ~6 km altitude with values
greater in the micro-physics run than in the two other configurations. In the three



configurations, high nebulosity values (greater than 0.8) are calculated close to the surface (0-
200 m), almost systematically in the two first configurations (operational and operational with
ice tuning). The AROME model tends to produce a sort of cloud residual in the planetary
boundary layer in the three configurations analyzed so the presence of a thin cloud close to
the surface cannot be ruled out. The thick-cloud episode during the warm and wet period of
2013 is well reproduced by ARPEGE together with the ice precipitation but the diamond dust
episode cannot be calculated during the cold and dry episode. No liquid water clouds are
estimated by the ARPEGE analyses.
Considering 5-day back trajectories from Dome C and global distributions of IWV over
the Antarctic in 2011 and 2013 tends to show that the thick-cloud episodes observed during
the warm and wet periods above Dome C can be attributed to air masses with an oceanic
origin whilst the diamond dust episode occurring during the cold and dry periods can be
attributed to air masses with continental origins. This is confirmed by the ARPEGE
temperature tendencies calculated during the warm and the cold periods of 2013 that are
mainly dominated by the advection components whilst the water vapour tendencies are
governed by both the advection and the microphysical processes.
The analysis of these two periods is going to be enlarged towards a climatological survey
of the presence of clouds and of their types above the Dome C station during the period 2009-
2016. We will combine measurements from the same instruments and outputs from the same
model together with  new instruments installed at the station providing the microphysical and
optical properties of the ice crystals that deposit at the surface.

**Acknowledgment**
The present research project HAMSTRAD programme (910) has been performed at the
Dome C station and was supported by the French Polar Intitute, Institut polaire français Paul-




Emile Victor (IPEV), the Institut National des Sciences de l'Univers (INSU)/Centre National
de la Recherche Scientifique (CNRS), Météo-France and the Centre National d'Etudes
Spatiales (CNES). The permanently manned Concordia station is jointly operated by IPEV
and the Italian Programma Nazionale Ricerche in Antartide (PNRA). We would like to thank
all the winterover personels who worked at Dome C on the different projects: HAMSTRAD,
Routine Meteorological Observations (RMO), aerosol Lidar and BSRN. Thanks to the British
Atmospheric Data Centre, which is part of the   Natural Environment Research Council
(NERC)  National Centre for Atmospheric Science (NCAS), for the calculation of trajectories
and access to European Centre for Medium-Range Weather Forecasts (ECMWF) data. We
have used NCEP Reanalysis data provided by the NOAA/OAR/ESRL PSD, Boulder,
Colorado, USA, from their Web site at http://www.esrl.noaa.gov/psd/. The authors also would
like to thank the CALIPSO science team for providing the CALIOP images at http://www-
calipso.larc.nasa.gov/.          HAMSTRAD         data       are       available       at
http://www.cnrm.meteo.fr/spip.php?article961&lang=en.   RMO    data    are    available    at
http://www.climantartide.it






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



**Figure Caption**


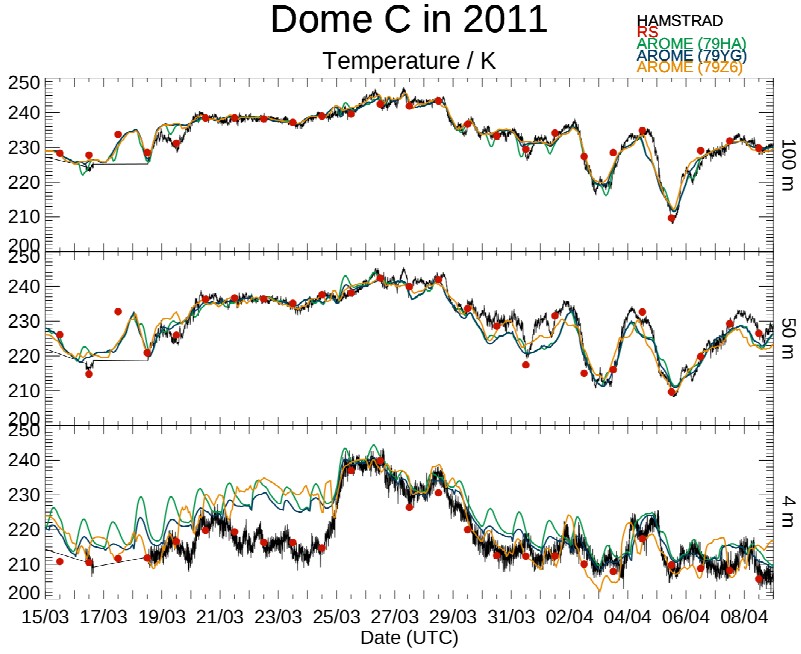


**Fig. 1:** From bottom to top: Time evolution of temperature from 15 March to 8 April 2011
above Dome C as measured by the HAMSTRAD radiometer (black line), the radiosondes (red
filled circles), and as calculated by the mesoscale model AROME according to different runs:
a) operational (green line), b) operational with ice tuning (blue line) and c) considering
ARPEGE micro-physics (orange line) at 4, 50 and 100 m.



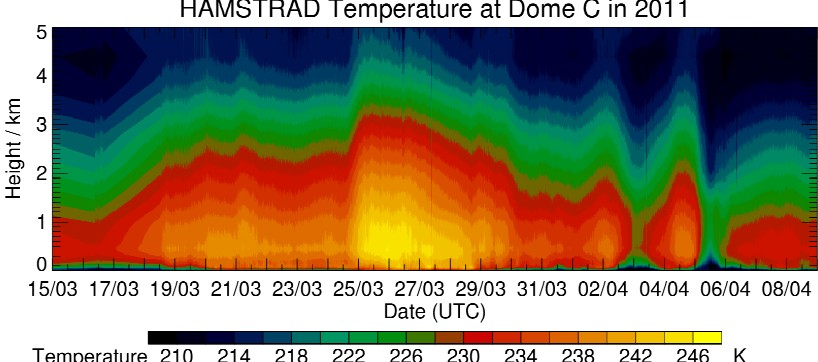

**Fig. 2:** Time evolution of temperature from 15 March to 8 April 2011 above Dome C as

measured by the HAMSTRAD radiometer from 0 to 5 km.





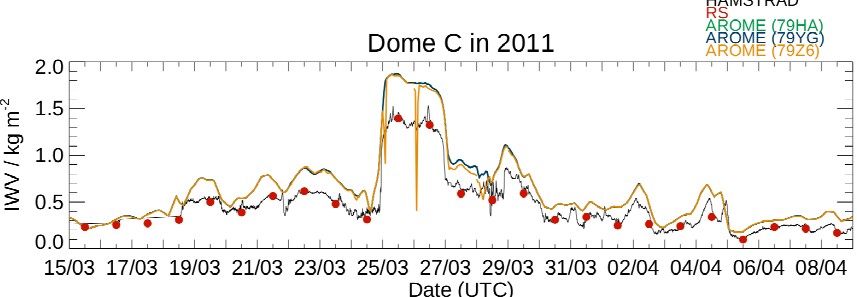

**Fig. 3:** Time evolution of IWV from 15 March to 8 April 2011 above Dome C as measured

by the HAMSTRAD radiometer (black line), the radiosondes (red filled circles), and as

calculated by the mesoscale model AROME according to different runs: a) operational (green

line), b) operational with ice tuning (blue line) and c) considering ARPEGE micro-physics

(orange line).





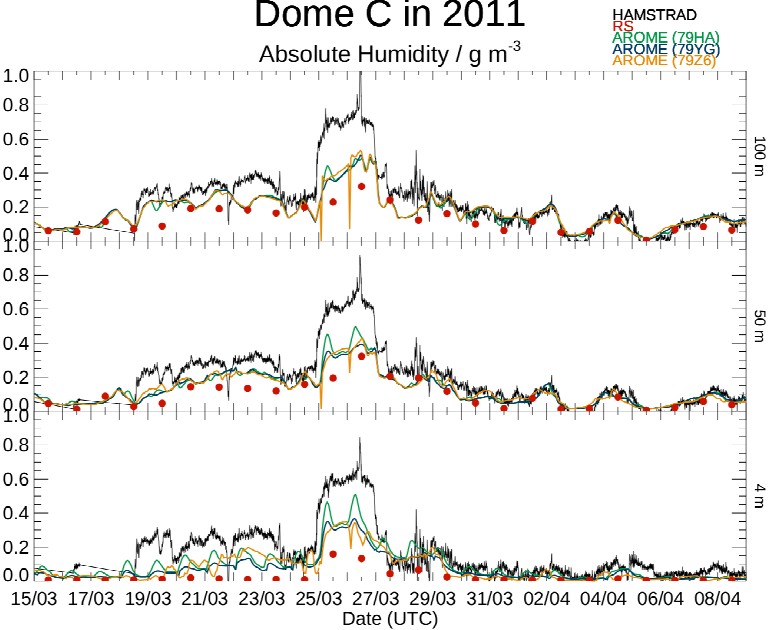

**Fig. 4:** From bottom to top: Time evolution of absolute humidity from 15 March to 8 April

2011 above Dome C as measured by the HAMSTRAD radiometer (black line), the

radiosondes (red filled circles), and as calculated by the mesoscale model AROME according

to different runs: a) operational (green line), b) operational with ice tuning (blue line) and c)

considering ARPEGE micro-physics (orange line) at 4, 50 and 100 m.




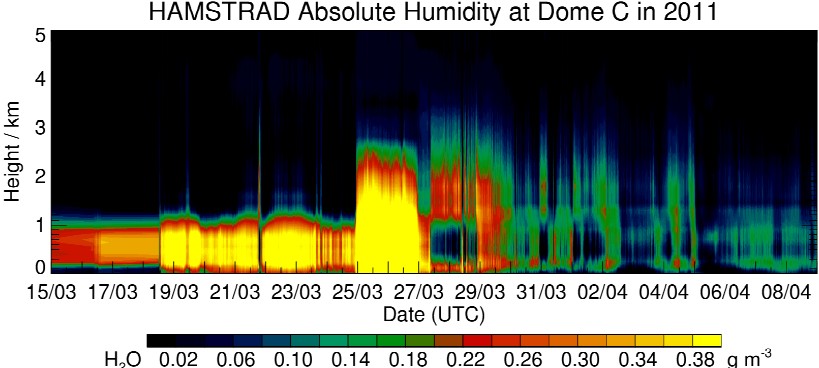


**Fig. 5:** Time evolution of absolute humidity from 15 March to 8 April 2011 above Dome C as

measured by the HAMSTRAD radiometer from 0 to 5 km.







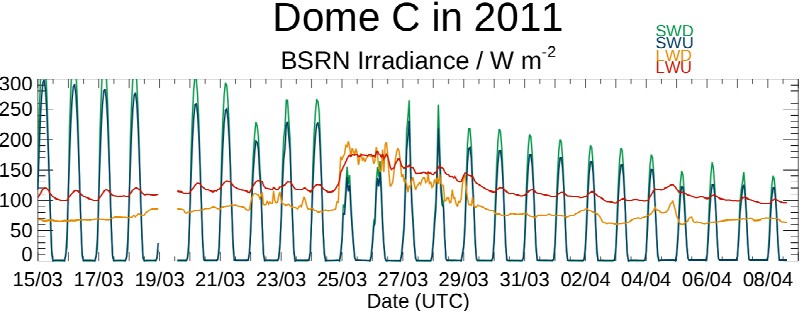

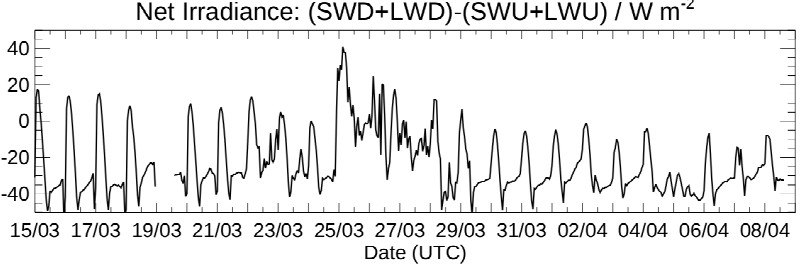

**Fig. 6:** (Top) Time evolution of downward shortwave radiation (SWD, green line), upward

shortwave radiation (SWU, blue line), downward longwave radiation (LWD, orange line),

and upward longwave radiation (LWU, red line) from 15 March to 8 April 2011 above Dome

C as measured by the BSRN instruments. (Bottom) Net irradiance (SWD+LWD-SWU-

LWU) as measured by the BSRN instruments.






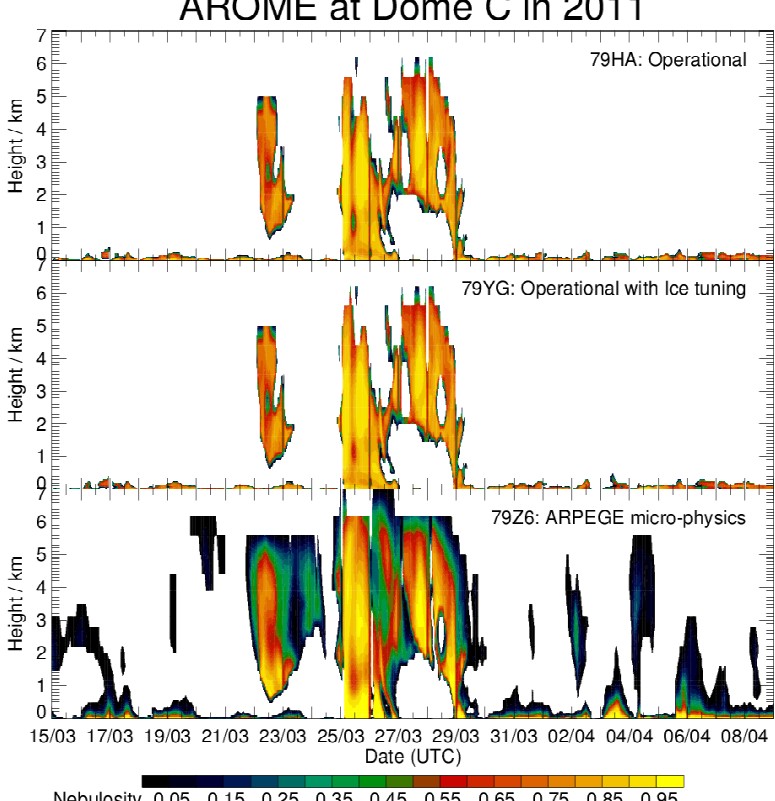

**Fig. 7:** (From top to bottom) Time evolution of nebulosity from 15 March to 8 April 2011
above Dome C as calculated by the mesoscale model AROME according to different runs:
operational (top), operational with ice tuning (center) and considering ARPEGE micro-
physics (bottom). See the text for further information regarding the AROME runs.




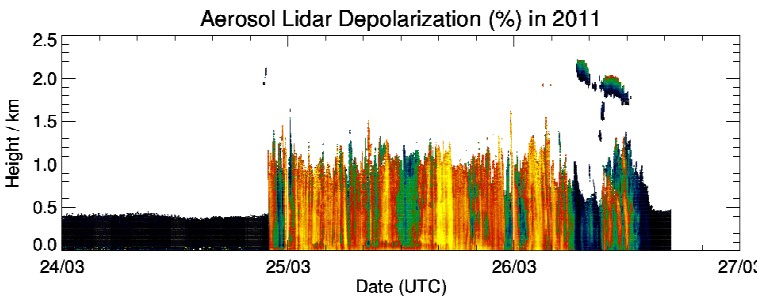

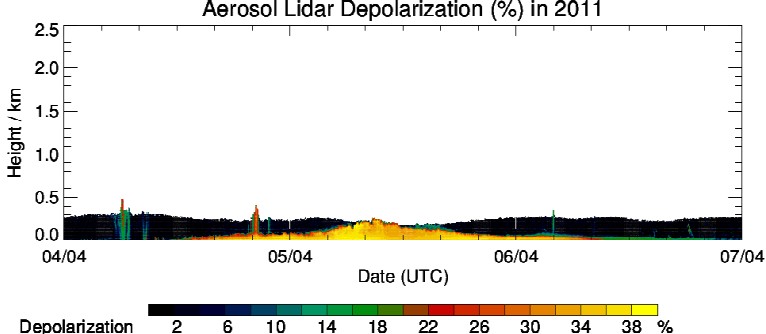

**Fig. 8:** (Top, from left to right) Aerosol depolarization as measured by the Lidar installed at
the Dome C station over the period 24-26 March 2011. (Bottom, from left to right) Aerosol
depolarization as measured by the Lidar installed at the Dome C station over the period 4-6
April 2011.




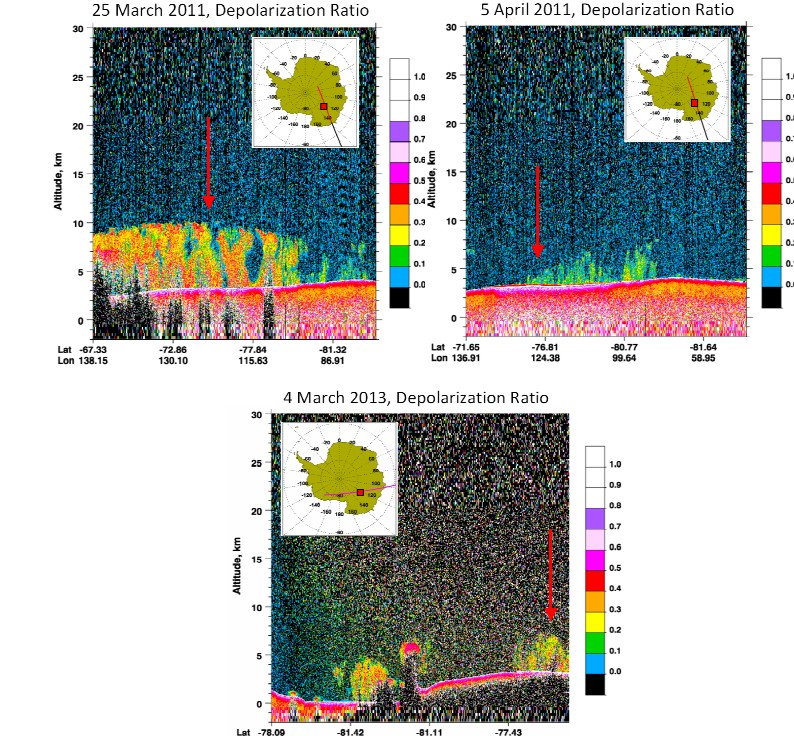



**Fig. 9:** Spaceborne Lidar CALIOP measurements of depolarization ratio along one orbit in the vicinity of the Dome C station on 25 March 2011 (top left), 5 April 2011 (top right) and 4 March 2013 (bottom). The red square represents the location of the Dome C station. The red vertical arrow represents the approximate location of the Dome C station.









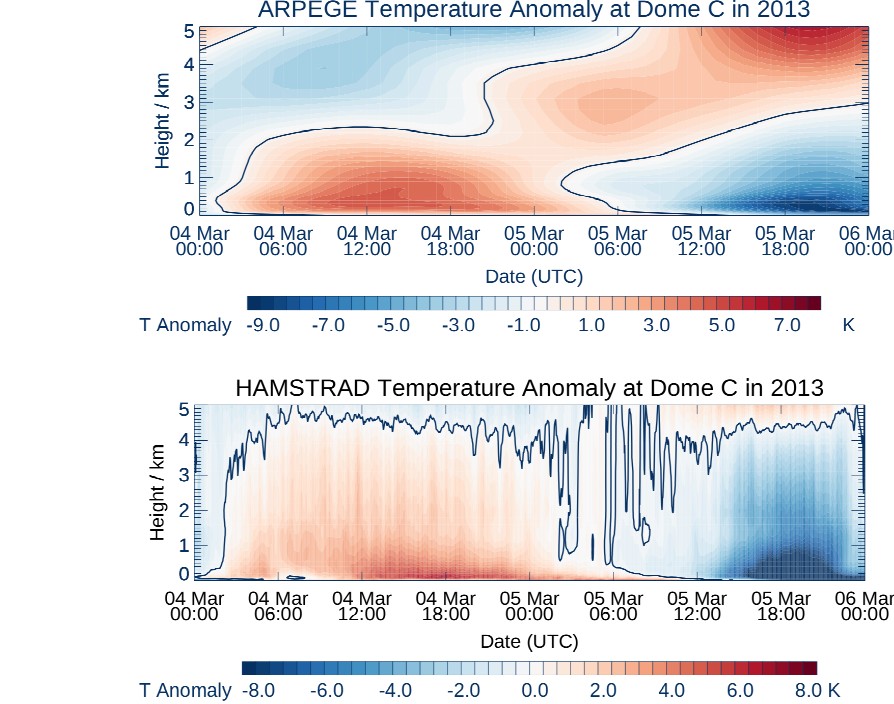

**Fig. 10:** Temperature anomaly from 4 to 5 March 2013 above the Dome C station as

calculated by the ARPEGE model (top) and as measured by the HAMSTRAD radiometer

(bottom).





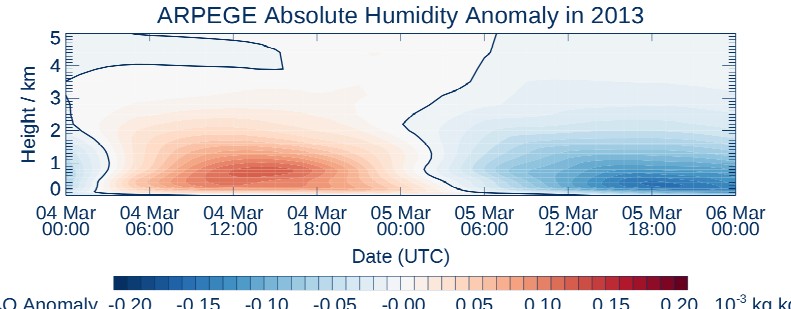


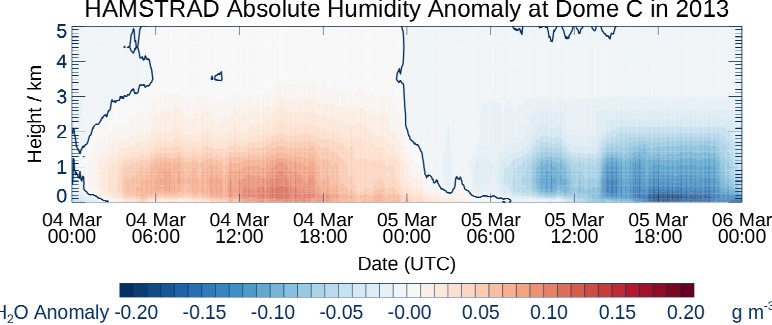


**Fig. 11:** Absolute Humidity anomaly from 4 to 5 March 2013 above the Dome C station as

calculated by the ARPEGE model (top) and as measured by the HAMSTRAD radiometer

(bottom).






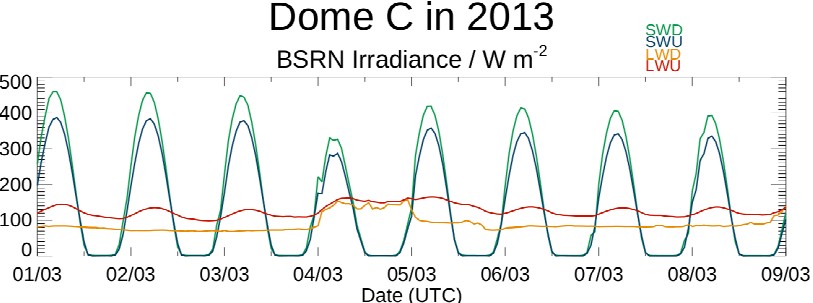

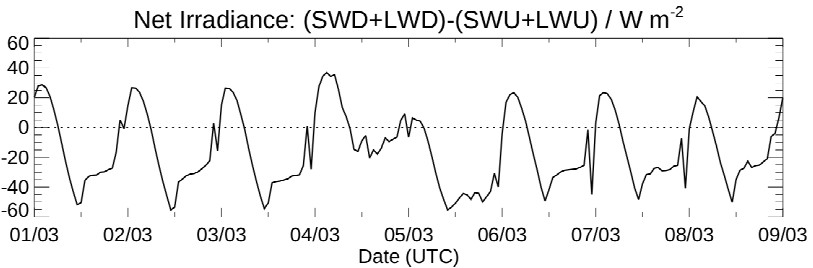

925

**Fig. 12:** (Top) Time evolution of downward shortwave radiation (SWD, green line), upward

shortwave radiation (SWU, blue line), downward longwave radiation (LWD, orange line),

and upward longwave radiation (LWU, red line) from 1 to 9 March 2013 above Dome C as

measured by the BSRN instruments. (Bottom) Net irradiance (SWD+LWD-SWU-LWU) as

measured by the BSRN instruments.

931

932

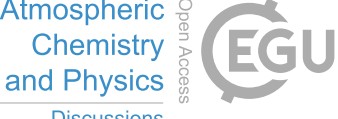


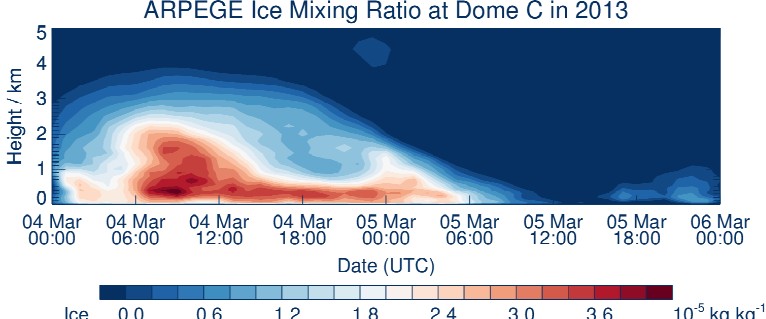


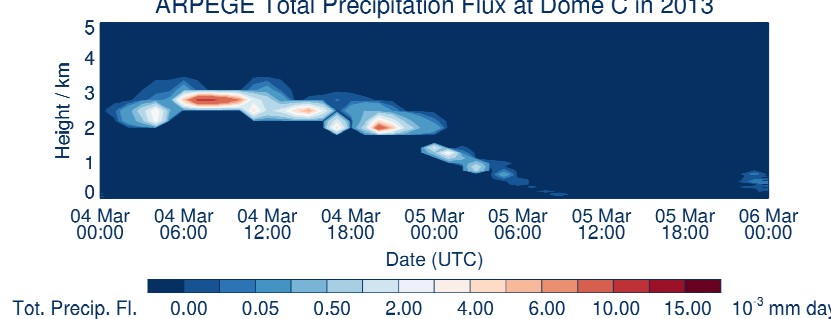


**Fig. 13:** Time evolution of the Ice Water Mixing ratio (top) and of the Total Precipitation

Flux (bottom) from 4 to 5 March 2013 above the Dome C station as calculated by the

ARPEGE model.







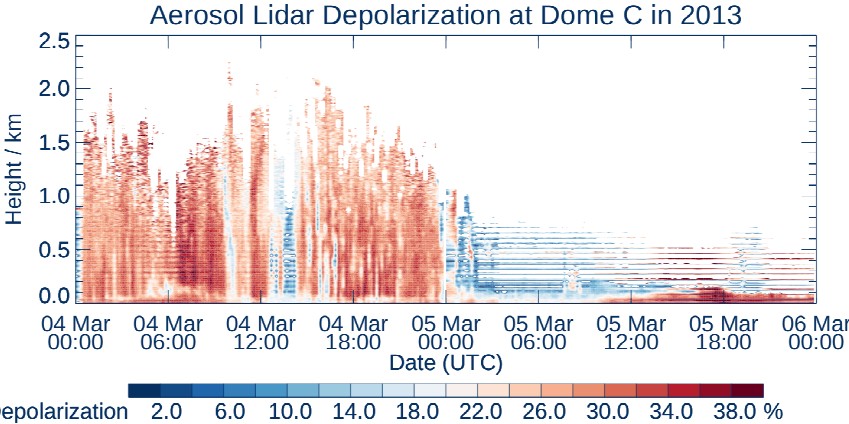

939

**Fig. 14:** Time evolution of the Depolarization ratio (%) from 4 to 5 March 2013 above the

Dome C station as measured by the aerosol Lidar installed at Dome C.

942






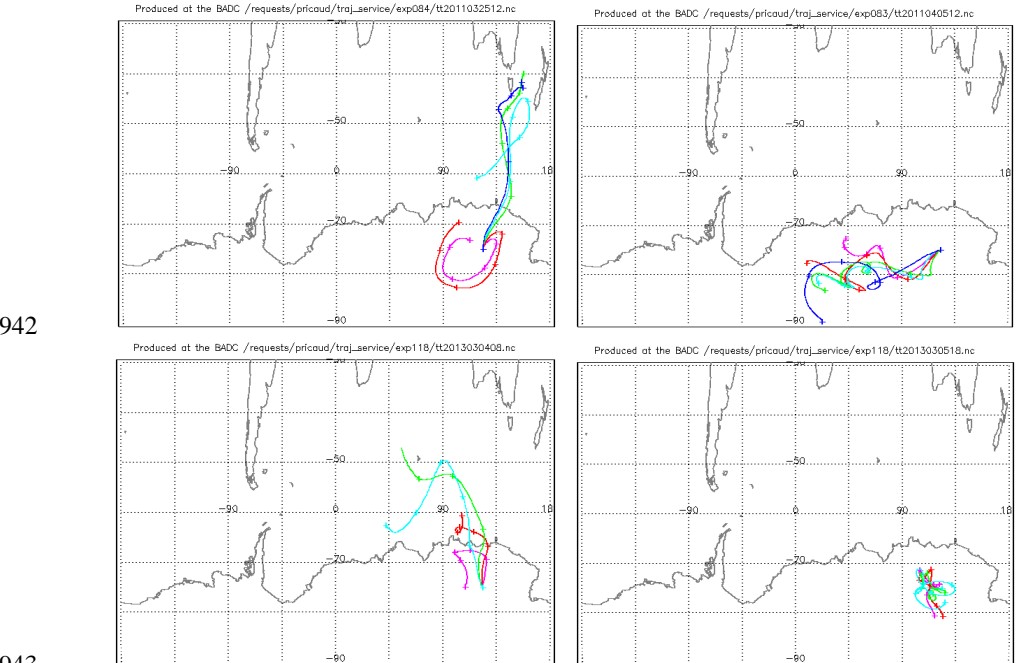


**Fig. 15:** (Top) Five-day back-trajectories of air masses originated from Dome C on 25 March

2011 at 12:00 UTC (left) and on 5 April 2011 at 12:00 UTC (right) at 650 (pink line), 600

(red line), 500 (green line), 400 (light blue line) and 300 hPa (dark blue line). (Bottom) Same

as top but on 4 March 2013 at 08:00 UTC (left) and on 5 March 2013 at 18:00 UTC (right).

951
952





952

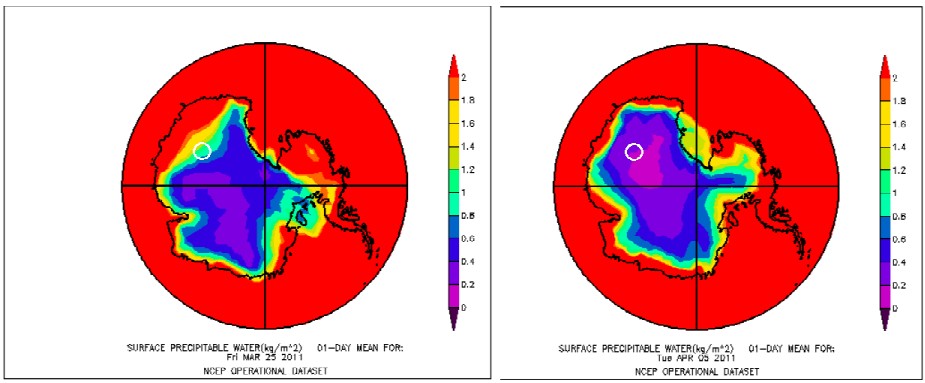

953

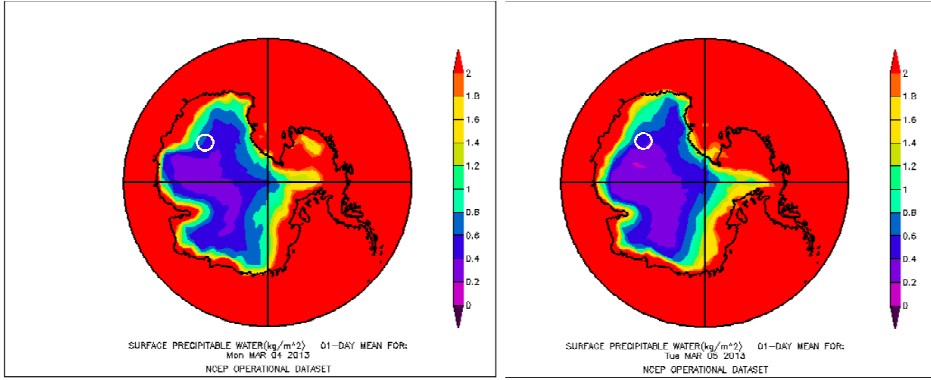

**Fig. 16:** IWV calculated above the Antarctic continent from the NCEP/NCAR operational

analyses on 25 March 2011 (top left), 5 April 2011 (top right), 4 March 2013 (bottom left)

and 5 March 2013 (bottom right). The white circle represents the position of the Dome C

station.

958

959



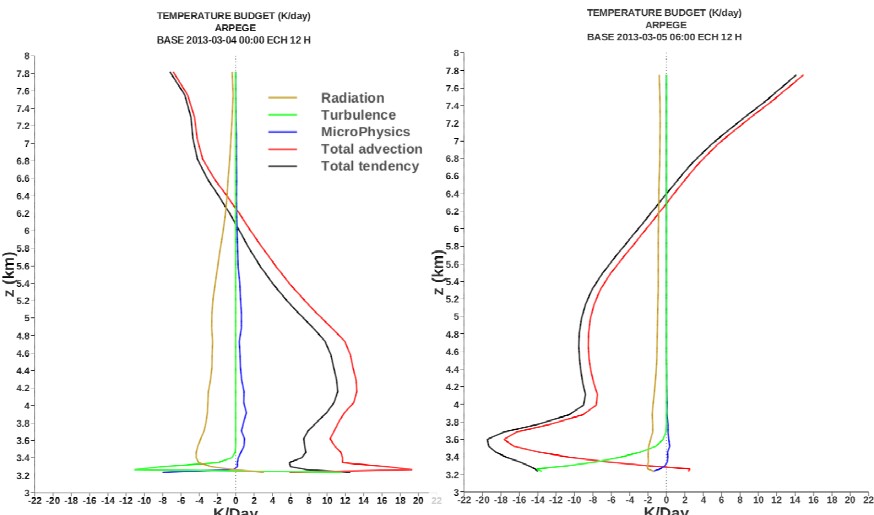


**Fig. 17:** (Left) Temperature budget calculated by ARPEGE on 4 March 2013 over the 12-h
period 00:00-12:00 UTC induced by radiation (brown), turbulence (green), microphysics
(blue), total advection (red) showing the total tendency (black). (Right) Same as Left but on 5
March 2013 over the 12-h period 06:00-18:00 UTC.




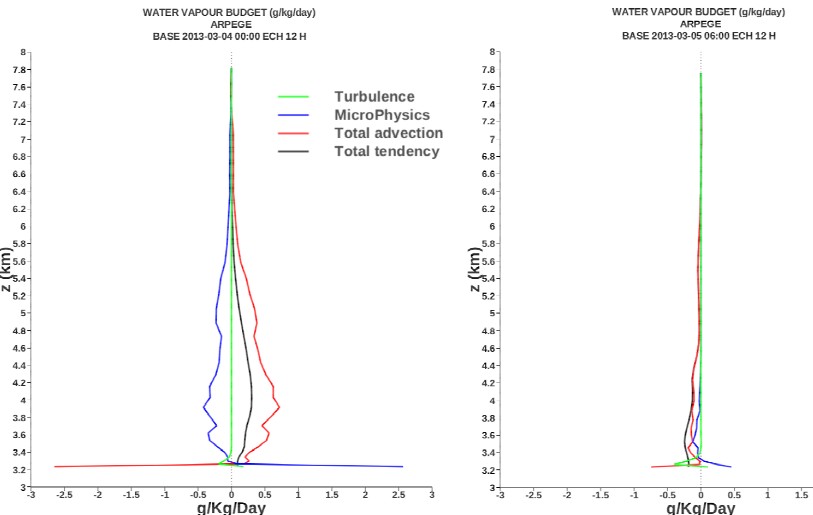

964

**Fig. 18:** (Left) Water vapour budget calculated by ARPEGE on 4 March 2013 over the 12-h

period 00:00-12:00 UTC induced by turbulence (green), microphysics (blue), total advection

(red) showing the total tendency (black). (Right) Same as Left but on 5 March 2013 over the

12-h period 06:00-18:00 UTC.

969

970