# Peer review of "Genesis of Diamond Dust and Thick Cloud Episodes observed"

_Atmospheric Chemistry and Physics, 2016_

## Referee Comment (RC1) · Anonymous Referee #1 · 17 Nov 2016

This manuscript intends to study of cold weather conditions (over Antarctica). It focuses on clouds and diamond dust, and various observational platforms and model simulations over more than 1 month of observations. There are several issues with this manuscript and need to be improved significantly before goes to publication.

Major/minor issues: 1. Objectives are not clearly set up 2. lots of information but nothing to do with objectives 3. diamond dust definition is not right 4. See Gultepe et al AMS Bulletin/Atmos Res for ice fog, also diamond dust definitions. DD is not suspending in the air but ice fog it does. DD has large particles and usually plates which shines as diamond. 5. Better to have results on 1) clouds and 2) DD, then fill up with your knowledge/observations. 6. What is the method here? we know that all these observations are important. How do you come up with conclusions? 7. I don't see clear conclusions???? 8. what are the issues with models? for these conditions?

9. manuscript should be reduced, using with tables and focusing with objectives 10. scientifically is a poor paper, no new ideas or relate objectives to new instrumental platforms or models.

Because of above I see that paper needs to be improved significantly before making a decisions if it is appropriate for this ACP.
* * *

---

## Referee Comment (RC2) · Anonymous Referee #2 · 18 Nov 2016

I find this paper to be a clear discussion of factors contributing to "warm" and "cold" clouds over Dome C. Also, it offers a good comparison of a number of measurement techniques and modelling. The result that advection patterns are a main driver of the differences between the warm and cold clouds is interesting, and I think it is suitable for ACP. The conclusions are mostly well founded. The paper is quite long, but it is very well organized and easy to read except for some small grammatical issues that I include with my following comments.

Comments

1) Line 20-23 – Re-write something like "Episodes of thick cloud and diamond dust during 15 March to 8 April 2011 and 4 to 5 March 2013 in the atmosphere above Dome C (Concordia station, Antarctica, 75°06'S, 123°21'E, 3233 m amsl) were measured

and modelled."

2) Line 23 – "The measurements were obtained from the following instruments: 1) . . ."

3) Line 35 – remove "by all datasets"

4) Line 76 – remove "also"

5) Lines 85-88 – ". . . experiences cloud about 30% of the time at altitudes below 3 km and less than 10% of the time above 5 km. Cloud occurrence over the western continental region is about 50% below 3 km and about 30% from the surface up to 8 km."

6) Lines 109-110 – replace "investigating" with "on". Remove "from" in both places. Replace period after "used" with a semi-colon.

7) Line 121 – please add a reference for the AROME model here.

8) Line 129 – remove first "the".

9) Line 132 – remove first "the". Remove third "the".

10) Lines 154-155 – Please clarify the biases. E.g. "Compared with the radiosondes, the radiometer temperatures are biased 1-5 K lower at altitudes below 4 km and 5-10 K higher above 4 km." Similarly for the "wet" bias.

11) Line 178 – remove first "the".

12) Lines 224-226 – "AROME was used within the GEWEX Atmospheric Boundary Layer Study 4 (GABLS4) to study the meteorological evolution over the Dome C station (Bosveld et al., 2014)."

13) Lines 269-271 – Can you offer any explanation for this large difference?

14) Lines 309-311 - AROME is wetter than the radiosondes in the figure. I think your respective values here should be reversed.

15) Line 316 – remove "much"

16) Line 322 – Change "Consistently" to "Consistent", replace "drawn with" with "concerning" and remove the last "the".

17) Line 340 – remove "much"

18) Line 347-348 – Change to "There was no abrupt increase of longwave downward radiation as during the warm and..."

19) Line 395 – Change to "We gain more insight into the vertical structure..."

20) Lines 423-424 – Please elaborate on why mixing of the air beneath the inversion causes the supersaturation w.r.t. ice.

21) Lines 433-435 – Change to "The second episode, which is much shorter than the first, relies on the same datasets presented in Section 3. The only difference is that the model analyses are only from the meteorological operational model ARPEGE that..."

22) Lines 443-444 – Please elaborate on "...with a explain transition propagating in the HAMSTRAD data up to 4 km altitude, probably due to the vertical resolution of the microwave measurements." Are you saying that the transition (and please define transition) reaches 4 km because the vertical resolution is poor? You mention that the resolution is 500 m. Does that mean the transition height might be 3.6 km instead of 4 km?

23) Line 474-475 – "Consequently, this 12 hour period on 5 March can neither be attributed to clear sky nor to thick cloud episodes."

24) Line 482-483 – Change "The ARPEGE simulation indicates an ice cloud from the surface to near 4 km on 4 March with the top altitude decreasing..."

25) Lines 494-495 - What is the temperature in this regime? You show no temperatures for episode 2, only tendencies. The reduced polarization may indicate water droplets, but it could also be due to change in the crystal ice habit. Your statement needs to be

supported by appropriate temperatures or changed to read that there is a discrepancy in the interpretation between the reduced polarization and the temperature.

26) Lines 496-498 – What vertical structure are you referring to? There is considerable vertical structure in the depolarization across the time period 1000-2400 on March 5 and within the lower few hundred metres. You can't say with any certainty here that there was no precipitation of ice crystals. Even diamond dust is large enough to fall out unless there is sufficient vertical wind to maintain their suspension. Please revise.

27) Line 510 – Please revise to "…of ice crystals with a longer suspension time in the air."

28) Lines 520-523 – change "were presented" on line 522 to "is discussed". You are discussing the "impact".

29) Line 564 – Replace "If we consider the" with "The". Remove "as".

30) Line 565 – Remove the first "the".

31) Line 566 – Replace ", we obviously remark" with "show".

32) Line 569 – Remove "indeed".

33) Line 575 – Replace "slight" with "smaller".

34) Line 578 – "thick-cloud episodes"

35) Line 580 – "… the diamond dust episodes occurred during…"

36) Line 584 – "Here, we attribute the tendencies of …"

37) Line 585 and 586 – Replace "into" by "among"

38) Section 5.3 – Please explain why this attribution was not be done for episode 1?

39) Line 604 – remove "a". change "on" to "in"

40) Line 607 – what are "small precipitations"?

41) Line 609-610 – Does this sentence, which refers to dehydration of the PBL and includes "precipitation", contradict your statement at the end of section 4.4 that the diamond dust does not precipitate? See also above comments 26 and 27.

42) Lines 600-610 – You refer to microphysics as the one of the factors influencing the water vapour budget. I know you are using the model with microphysics, but it is not clear from your discussion if the influence on the water budget is truly microphysics or just the presence of cloud. Please clarify in Section 5.3.

43) Line 650 – "Since both downward and upward longwave radiation are greater than..."

44) Line 665-666 – "... suggesting that the cloud consisted of smaller ice crystals that may remain suspended in the air longer."

---

## Author Comment (AC1) · 6 Feb 2017

This manuscript intends to study of cold weather conditions (over Antarctica). It focuses on clouds and diamond dust, and various observational platforms and model simulations over more than 1 month of observations. There are several issues with this manuscript and need to be improved significantly before goes to publication.

Because of above I see that paper needs to be improved significantly before making a decision if it is appropriate for this ACP.

→ Specific changes have been made in response to the reviewers' comments and are described below.

Major/minor issues:

1. Objectives are not clearly set up. Lots of information but nothing to do with objectives.
→ We have clarified this crucial point. The objectives of the paper are mainly to investigate the processes that cause the presence of thick cloud and diamond dust/ice fog episodes above the Dome C station based on observations and verify whether operational models can evaluate them. The title has been modified accordingly into "Genesis of Diamond Dust, Ice Fog and Thick Cloud Episodes observed and modelled above Dome C, Antarctica"

2. Diamond dust definition is not right. See Gultepe et al. AMS Bulletin/Atmos Res for ice fog, also diamond dust definitions. DD is not suspending in the air but ice fog it does. DD has large particles and usually plates which shines as diamond.
→ Consistent to the definitions listed in Gultepe et al. (2014), we present below the definition of "diamond dust" and "ice fog" taken from the National Snow and Ice Data Center (NSIDC) on their site *http://nsidc.org/cryosphere/glossary*:

**Diamond dust**: a type of precipitation composed of slowly falling, very small, unbranched crystals which often seem to float in the air; it may fall from a high cloud or from a cloudless sky, it usually occurs under frosty weather conditions (under very low air temperatures).

**Ice fog**: a suspension of numerous minute ice crystals in the air, reducing visibility at the earth's surface; the crystals often glitter in the sunshine; ice fog produces optical phenomena such as luminous pillars and small haloes.

From Girard and Blanchet (2001), ice fog is distinguished from diamond dust by the high concentration of ice crystal of smaller diameters. Ice fog, ice crystals are generally closer to spherical shape and their number concentration exceeds $1000 \text{ L}^{-1}$ while their mean diameter is below 30 µm. From Walden et al. (2003), the atmospheric ice crystals over the Antarctic Plateau in winter is mainly constituted of three major types: diamond dust, blowing snow and snow grains. When sorted by number, Lawson et al. (2006) attribute 30% of the crystals recorded at the South Pole Station to rosette shaped (mixed-habit rosettes, platelike polycrystals, and rosette shapes with side planes), 45% to diamond dust (columns, thick plates, and plates), and 25% to irregular. By mass, the percentages are 57% rosette shapes, 23% diamond dust, and 20% irregular. In conclusion, in the literature of the ice crystals over Antarctica and particularly over the Antarctic Plateau, there is little mention of ice fog except some studies performed in the coastal areas of e.g. McMurdo in Lazzara (2010). Based on Gultepe et al. (2015) and Arctic

studies, the maximum size for ice fog crystals is about 200 μm with diamond dust ice crystal sizes greater than 200 μm.

We have thus modified the paragraph in L. 418 related to the diamond dust and inserted a discussion on ice fog.

At Dome C, in 2013, an ICE-CAMERA was installed by Dr M. del Guasta on the roof of the shelter where both the HAMSTRAD and the aerosol Lidar were set up. This camera was able to take on an hourly rate a picture of the ice crystal grains deposited at the surface of the camera.

The ICE-CAMERA (http://lidarmax.altervista.org/englidar/_Antarctic%20Precipitation.php) is equivalent to a flatbed scanner. It was built in order to operate unattended in Polar Regions for the study of precipitating ice grains. Precipitation is collected on a glass plate, where it is photographed with 5 um resolution hourly. After each scan, the glass plate is electrically-heated in order to sublimate the ice grains. Image-processing software is used for the automatic characterization and counting of grains. A sub-section has been inserted in the new version of the manuscript.

```
2.6. The ICE-CAMERA
At Dome C, in 2013, an ICE-CAMERA was installed on the roof
of the shelter where both the HAMSTRAD and the aerosol Lidar
were set up. This camera was able to take on an hourly rate
a picture of the ice crystal grains deposited at the surface
of        the        camera.        The        ICE-CAMERA
(http://lidarmax.altervista.org/englidar/_Antarctic%20Precip
itation.php) is equivalent to a flatbed scanner. It was built
in order to operate unattended in Polar Regions for the study
of precipitating ice grains. Precipitation is collected on a
glass plate, where it is photographed with 5 um resolution
hourly. After each scan, the glass plate is electrically-
heated in order to sublimate the ice grains. Image-processing
software is used for the automatic characterization and
counting of grains.
```

The distinction between ice-fog and diamond dust is quite recent. In the past, the two phenomena were completely confused at least in Antarctica. We often have "frozen low clouds/fogs" in Concordia showing little evident precipitation on the glass plate of ICE-CAMERA, and other cases with bigger ice crystals. Of course particles from (high) clouds are bigger, while in-situ formed ones are smaller, and this, regardless of their classification into ice-fog and/or diamond dust. In fact, the use of the Lidar instrument actually shows the region of formation, a point indirectly assessed in the ref. papers suggested by the referees, and under this point of view, we can distinguish between cloud-originated particles (precipitation) and locally formed particles (ice fog and diamond dust). In the ICE CAMERA, very small particles (less than 20 um diameter) are not detected nor counted. About the form of the crystals, for focusing reasons before 2014, their appearance is sub spherical.

Figure R1 shows, as an example for episode 2 in 2013, the warm period on 4 March and the beginning of the cold period of 5 March: a) 4 March at 12:31 UTC; b) 4 March at 18:31 UTC; c) 5 March at 00:31 UTC; d) 5 March at 06:31 UTC and e) 5 March at 09:31 UTC. The 1-mm scale is indicated on each frame. The camera stopped functioning after 5 March at 09:31 UTC. From the images taken on 4 and early on 5 March, we can see that crystals are mainly constituted of elongated columns and/or plates, at least the bigger ones. From Figure R1, the

size of the bigger crystals is about 300-400 um on 4 March and about 400-600 um on 5 March. From the literature, it seems that on 4 March, the size and form of the crystals are consistent with blowing snow. On 5 March at 09:30 UTC, we are slightly before the period of diamond dust that is to say after 12:00 UTC. On 5 March, the crystals seem to be bigger in size than on 4 March, elongated, but far much bigger that they should be from literature if they were only labelled as diamond dust but again we are few hours before the diamond dust period. Note the differences in the detection of ice grains due to the focus device present on 13 March 2011 (Fig. R2) and absent on 4 March 2013 (Fig. R1).

[Figure]

[Figure]

*Figure R1. (From top to bottom and from left to right) Pictures taken from the ICE-CAMERA installed on the roof of the shelter where the HAMSTRAD and the aerosol Lidar are installed showing grains of ice crystals deposited over one hour on the surface of the camera in 2013 on: a) 4 March at 12:31 UTC; b) 4 March at 18:31 UTC; c) 5 March at 00:31 UTC; d) 5 March at 06:31 UTC and e) 5 March at 09:31 UTC. The 1-mm scale is indicated on each frame.*

[Figure]

*Figure R2. Example of a picture taken from the ICE-CAMERA on 13 March 2012 when the autofocus was active.*

We also present the histograms for the "lengths" (major axis of the ellipsoid fitting the particle shape) for 4 and 5 March 2013 (Fig. R3). Until 2014, the image processing was relatively inadequate for the absence of autofocus, and also for the excess of heating of the collecting

plate. The device was replaced with the actual one on 2014. As a result the 2013 histograms show always small particles with comparison with what observed later on.

Two histograms of particle lengths for the whole March 2013 and the whole March 2015 are also shown in Figure R4. The crystal lengths in 2015 are apparently much longer than in 2013. For this reason, as the "threshold" size of particles for fog ice is approx. 200 μm (following the literature suggested by the referees) for 5 March 2013, we would be on the "ice fog" side, rather than the diamond dust side. But this choice would be caused by an underestimation of the particle size resulting when following literally the ICE-CAMERA data of that period. As explained earlier, data recorded in 2013 are not quantitatively reliable until 2014.

[Figure]

*Figure R3. Ice grain size distribution measured on 4 March (left) and on 5 March (right) 2013 at the Dome C station by the ICE CAMERA.*

[Figure]

*Figure R4. Ice grain size distribution measured in March 2013 (left) and in March 2015 (right) at the Dome C station by the ICE CAMERA.*

When looking at the ICE-CAMERA photos for several years, plates are relatively scarce in all the observed Concordia crystals. Also ice rosettes, columns, etc. shine in the sun like diamonds, and the diamond dust brilliancy is not due to plates only. This is just atmospheric optics. In addition, Gultepe et al. (2014) worked in the Arctic, whilst our study refers to the Antarctic in Concordia.

In conclusion, we cannot confirm or infirm that the low level ice crystals observed at Dome C are diamond dust or ice fog. So, we prefer keeping these two terms in the title and in the discussion of the revised manuscript.

References:

Girard, E. and Blanchet, J. P.: Microphysical parameterization of Arctic diamond dust, ice fog, and thin stratus for climate models. J. Atmos. Sci., 58, 1181-1198, 2001.

Gultepe, I., Kuhn, T., Pavolonis, M., Calvert, C., Gurka, J., Heymsfield, A. J., Liu, P. S. K., Zhou, B., Ware, R., Ferrier, B., Milbrandt, J. and Berstein, B.: Ice fog in Arctic during FRAM-ICE Fog project: Aviation and Nowcasting Applications, Bull. Am. Meteor. Soc., 95, 211-226, 2014.

Lawson, R. P., Baker, B. A., Zmarzly, P., O'Connor, D., Mo, Q., Gayet, J.-F. and Shcherbakov, V.: Microphysical and optical properties of atmospheric ice crystals at South Pole Station, J. Appl. Meteor. Climatol., 45, 1505-1524, 2006.

Lazzara, M.: Diagnosing Antarctic fog, 5[th] International Conference on Fog, Fog Collection and Dew, Münster, Germany, 25-30 July 2010. Vol. 1, p. 150. (http://meetingorganizer.copernicus.org/FOGDEW2010/FOGDEW2010-150.pdf)

Walden, V. P., Waren, S. G. and Tuttle, E.: Atmospheric ice crystals over the Antarctic Plateau in winter, J. Appl. Meteor., 42, 1391-1405, 2003.

3. Better to have results on 1) clouds and 2) DD, then fill up with your knowledge/observations. $\rightarrow$ We have 2 distinct periods in 2011 and 2013 covered by 2 different operational models showing both thick clouds and diamond dust/ice fog episodes. We investigate the processes that drive these observed episodes and check whether the operational models can or cannot detect them.

We have entirely restructured the revised manuscript focussing on clouds and diamond dust/ice fog based on observations, then evaluated by the operational models, followed by the evolution of meteorological parameters (radiation, temperature and water vapour) during these episodes. We finally discuss the processes that contribute to the presence of these episodes: origin of air masses, integrated water vapour and temperature/water vapour budgets.

Figure 12 (Figure R5) have been modified in order to highlight the impact of the presence of the thick cloud and of the diamond dust/ice fog episodes on the net irradiances. Over the period 1-8 March 2013, considering clear sky days (excluding 4 and 5 March), the net irradiance between 14:00 and 17:00 UTC is $-30.55$ W m$^{-2}$ whilst on 4 March (thick cloud episode) the net irradiance has increased to $-16.75$ W m$^{-2}$ and on 5 March (diamond dust/ice fog episode) the net irradiance has decreased to $-45.52$ W m$^{-2}$.

[Figure]

[Figure]

*Figure R5: (Top) Time evolution of downward shortwave radiation (SWD, green line), upward shortwave radiation (SWU, blue line), downward longwave radiation (LWD, orange line), and upward longwave radiation (LWU, red line) from 1 to 9 March 2013 above Dome C as measured by the BSRN instruments. (Bottom) Net irradiance (SWD+LWD-SWU-LWU) as measured by the BSRN instruments. The horizontal green dashed line represents the net irradiance averaged between 14:00 and 17:00 UTC (represented by green diamonds) from 1 to 8 March excluding 4 and 5 March. The red and blue diamonds represent the net irradiances measured between 14:00 and 17:00 UTC on 4 March (thick cloud episode) and 5 March (diamond dust/ice fog episode), respectively.*

Figure 12 has been replaced by Figure R5 in the new version of the manuscript.

4. What is the method here? We know that all these observations are important. How do you come up with conclusions? I don't see clear conclusions???? What are the issues with models? for these conditions?

→ In general, in the lower troposphere, ARPEGE and HAMSTRAD temperature datasets are very consistent to each other (see Fig. 10 of the first submitted manuscript). If we now consider into detail the evolution of the vertical temperature structures during the 2 episodes, using also radiosondes measurements at 12:00 UTC, interesting conclusions can be deduced from Figure R6. During the warm episode (thick cloud) on 4 March, there is a sharp positive vertical temperature gradient at 12:00 and 18:00 UTC within the first 100-200 m altitude layer in the planetary boundary layer from a very cold surface of about 230 K reaching a maximum of about 240-242 K. Above, the atmosphere is rather isothermal and starts cooling around 400-800 m. All the datasets are very consistent to each other. During the cold episode on 5 March, the lower troposphere is generally colder that during the warm episode, but at 12:00 UTC there is no such a stable planetary boundary layer as observed a day before since the vertical gradient is only of a 3 K over 500 m altitude instead of 20 K over 200 m during the warm episode. Radiosonde, ARPEGE and HAMSTRAD profiles are consistent to each other. At 18:00 UTC, in the core of the diamond dust/ice fog episode, the vertical structure of the temperature observed by

HAMSTRAD showing a positive gradient (218 to 227 K from 100 to 500 m) is opposite to the negative gradient of ARPEGE temperature (228 to 226 K from the surface to 400 m). Consistent with Figure 12 (Figure R6), the radiative impact of the thick cloud is to enlarge the net irradiance by about 15 W m$^{-2}$ thus to locally increase temperature as measured by HAMSTRAD and calculated by ARPEGE whilst the radiative impact of the diamond dust/ice fog is to reduce the net irradiance by about 15 W m$^{-2}$ thus to locally decrease temperature as measured by HAMSTRAD but not calculated by ARPEGE.

[Figure]

*Figure R6:* *Vertical distribution of temperature measured by HAMSTRAD (solid line), and radiosondes (dashed lines) and calculated by ARPEGE (dotted lines) on 4 March 12:00 UTC (red line) and 18:00 UTC (orange line) and on 5 March 12:00 UTC (blue line) and 18:00 UTC (green line). Note radiosondes are only available at 12:00 UTC.*

Figure R6 has been inserted in the new version of the manuscript together with this new paragraph in section 4.4.

We have inserted this new paragraph as a synthesis together with this new reference in section 4.4:

> ```
> The operational models take into account interactions between
> liquid and solid water phases but are enable to actually
> simulate the number of droplets that depends on their sizes.
> Consequently, models can estimate the presence of thick clouds
> but cannot reproduce diamond dust/ice fog episodes. A more
> sophisticated cloud microphysics such as a two-moment scheme
> as LIMA (Liquid Ice Multiple Aerosols) scheme (Vié et al.,
> 2016) and an explicit aerosol scheme (Girard et al., 2001)
> would favour the local production of ice crystals in the
> planetary boundary layer.
> ```

Vié, B., Pinty, J. P., Berthet, S., & Leriche, M. : LIMA (v1. 0): A quasi two-moment microphysical scheme driven by a multimodal population of cloud condensation and ice freezing nuclei. Geosci. Model Dev., 9, 567-586, 2016.

7. Manuscript should be reduced, using with tables and focusing with objectives

→ The manuscript has been considerably shortened (1 page) and reorganized (see point 3). 3 Figures have been removed (Figs. 1, 3 and 4), 2 new Figures have been inserted (Figs. R6 and R7), and 1 Figure has been upgraded (Fig. 12 or Fig. R5). 5 new references have been inserted.

8. Scientifically is a poor paper, no new ideas or relate objectives to new instrumental platforms or models.

→ The new version of the manuscript replies to the main concerns of the reviewer.

---

## Author Comment (AC2) · 6 Feb 2017

I find this paper to be a clear discussion of factors contributing to "warm" and "cold" clouds over Dome C. Also, it offers a good comparison of a number of measurement techniques and modelling. The result that advection patterns are a main driver of the differences between the warm and cold clouds is interesting, and I think it is suitable for ACP. The conclusions are mostly well founded. The paper is quite long, but it is very well organized and easy to read except for some small grammatical issues that I include with my following comments.

→ We first would like to thank the reviewer for his/her positive general review. Specific changes have been made in response to his/her specific comments that are described below.

Comments:

1) Line 20-23 – Re-write something like "Episodes of thick cloud and diamond dust during 15 March to 8 April 2011 and 4 to 5 March 2013 in the atmosphere above Dome C (Concordia station, Antarctica, 75 ∘ 06'S, 123 ∘ 21'E, 3233 m amsl) were measured and modelled."
→ Done.

2) Line 23 – "The measurements were obtained from the following instruments: 1) . . ."
→ Done.

3) Line 35 – remove "by all datasets"
→ Done.

4) Line 76 – remove "also"
→ Done.

5) Lines 85-88 – ". . . experiences cloud about 30% of the time at altitudes below 3 km and less than 10% of the time above 5 km. Cloud occurrence over the western continental region is about 50% below 3 km and about 30% from the surface up to 8 km."
→ Done.

6) Lines 109-110 – replace "investigating" with "on". Remove "from" in both places. Replace period after "used" with a semi-colon.
→ Done.

7) Line 121 – please add a reference for the AROME model here.
→ We have inserted the reference to Seity et al. (2011) here instead of in the AROME section 2.6. as requested by the reviewer.

Note that in the AROME section 2.7 we have replaced the reference Bosveld et al. (2014), relevant to GABLS3, by Bazile et al. (2015) that was more appropriate to GABLS4.

Bazile, E., Couvreux, F., Le Moigne, P., and Genthon, C.: First Workshop on the GABLS-4 Intercomparison, GEWEX Newsletter, pp18-19, August 2015.

8) Line 129 – remove first "the".
→ Done.

9) Line 132 – remove first "the". Remove third "the".
→ Done.

10) Lines 154-155 – Please clarify the biases. E.g. "Compared with the radiosondes, the radiometer temperatures are biased 1-5 K lower at altitudes below 4 km and 5-10 K higher above 4 km." Similarly for the "wet" bias.
→ Based on previous studies and comparisons with radiosondes and space-borne measurements reported in Ricaud et al. (2015), "we can infer three recommendations regarding the HAMSTRAD data. 1) HAMSTRAD IWV measurements from 2009 to 2014 are of excellent quality (linear Pearson correlation coefficients $r > 0.98$) and can be used without retrieving any bias. These 7-minute time resolution data can be suitable for any scientific analysis considering both an absolute comparison (in unit kg m$^{-2}$) and a relative time evolution of this parameter (e.g. a temporal anomaly). 2) HAMSTRAD temperature measurements are suitable for scientific analyses over the range 0-10 km with a high time correlation ($r > 0.80$) with radiosondes. The time evolution of this parameter over the period 2009-2014 with a resolution of 7 minutes is meaningful. Nevertheless, the vertical distribution of temperature from 0 to 10 km is subject to biases that need to be removed if the scientific analyses require the use of vertical profiling. 3) HAMSTRAD absolute humidity measurements are suitable for scientific analyses over the range 0-4 km. Above 4 km, both the amount of $H_2O$ dramatically decreases and the instrument loses sensitivity. The time evolution of this parameter over the period 2009-2014 and over the range 0-4 km with a resolution of 7 minutes is meaningful ($r > 0.70$). Nevertheless, the vertical distribution of absolute humidity from 0 to 4 km is subject to biases that need to be removed if the scientific analyses require the use of vertical profiling."

In the revised version, in order to shorten the manuscript, we no longer show IWV HAMSTRAD data and we do not compare absolute humidity HAMSTRAD data with radiosondes anymore. We have thus modified the incriminated sentences accordingly.

```
Statistically, it has been shown that, against radiosondes
from 2009 to 2014, there is a 1-5 K cold bias below 4 km, and
a 5-10 K warm bias above, with a high time correlation (linear
Pearson correlation coefficient r > 0.80).  There is a wet
bias of 0.1-0.3 g m⁻³ below about 2 km and a dry bias of ~0.1
g m⁻³ above, with a high time correlation below 4 km (r >
0.70). Yearly-averaged vertical profiles of the biases in
temperature and absolute humidity are also provided from 2009
to date at the same http address. Note we have not debiased
HAMSTRAD data in the present study.
```

11) Line 178 – remove first "the".
→ Done.

12) Lines 224-226 – "AROME was used within the GEWEX Atmospheric Boundary Layer Study 4 (GABLS4) to study the meteorological evolution over the Dome C station (Bosveld et al., 2014)."
→ Done.

13) Lines 269-271 – Can you offer any explanation for this large difference?
→ We no longer show a comparison between HAMSTRAD and AROME temperature in a planetary boundary layer in the revised manuscript. This line and the associated Figure have been removed.

14) Lines 309-311 - AROME is wetter than the radiosondes in the figure. I think your respective values here should be reversed.
→ Done.

15) Line 316 – remove "much"
→ Done.

16) Line 322 – Change "Consistently" to "Consistent", replace "drawn with" with "concerning" and remove the last "the".
→ Done.

17) Line 340 – remove "much"
→ Done.

18) Line 347-348 – Change to "There was no abrupt increase of longwave downward radiation as during the warm and. . ."
→ Done.

19) Line 395 – Change to "We gain more insight into the vertical structure. . ."
→ Done.

20) Lines 423-424 – Please elaborate on why mixing of the air beneath the inversion causes the supersaturation w.r.t. ice.
→ We have rewritten the incriminated sentence into:

```
In the Eastern Antarctic Plateau over all the seasons except
summer, a strong surface-based temperature inversion persists
in which small ice crystals referred to as diamond dust/ice
fog form in the boundary layer (Walden et al., 2003).
```

21) Lines 433-435 – Change to "The second episode, which is much shorter than the first, relies on the same datasets presented in Section 3. The only difference is that the model analyses are only from the meteorological operational model ARPEGE that. . ."
→ Done.

22) Lines 443-444 – Please elaborate on ". . .with a explain transition propagating in the HAMSTRAD data up to 4 km altitude, probably due to the vertical resolution of the microwave measurements." Are you saying that the transition (and please define transition) reaches 4 km because the vertical resolution is poor? You mention that the resolution is 500 m. Does that mean the transition height might be 3.6 km instead of 4 km?
→ The reviewer is right. Due to the vertical resolution of the microwave radiometer a "transition height" of 4.0 km should be labelled as 4.0 ± 0.5 km. The "transition region" we were referring to was the 0-K temperature anomaly isoline delimiting the warm period on 4 March to the cold period of 5 March in the lower troposphere. Along the vertical, the 0-K temperature anomaly isoline is located around 2.4 km on 4 March in the ARPEGE dataset whilst it is higher up around 4.0 ± 0.5 km in the HAMSTRAD data set. We have rephrased the incriminated sentence in the new version of the manuscript.

```
Considering  the  0-K  temperature  anomaly  isoline,  both
datasets  show  a  warm  period  on  4  March  followed  by  a  cold
period  on  5  March  from  the  surface  to  about  2.4  km  altitude
on  4  March  by  ARPEGE  and  to  about  4.0  ±  0.5  km  by  HAMSTRAD.
The  1.6-km  difference  between  the  altitude  of  the  0-K
temperature  anomaly  isoline  in  HAMSTRAD  and  in  ARPEGE  data
sets  on  4  March  might  be  due  to  the  poor  vertical  resolution
of  HAMSTRAD  compared  to  ARPEGE.
```

23) Line 474-475 – "Consequently, this 12 hour period on 5 March can neither be attributed to clear sky nor to thick cloud episodes."
→ Done.

24) Line 482-483 – Change "The ARPEGE simulation indicates an ice cloud from the surface to near 4 km on 4 March with the top altitude decreasing. . ."
→ Done.

25) Lines 494-495 - What is the temperature in this regime? You show no temperatures for episode 2, only tendencies. The reduced polarization may indicate water droplets, but it could also be due to change in the crystal ice habit. Your statement needs to be supported by appropriate temperatures or changed to read that there is a discrepancy in the interpretation between the reduced polarization and the temperature.
→ Both temperature and absolute humidity anomalies decrease over the period 00:00-10:00 UTC on 5 March (Figs. 10 and 11, respectively). We now can consider the time evolution of the number of ice crystals measured by the ICE CAMERA integrated over one hour from 4 to 5 March 2013 above the Dome C station (Fig. R7). We notice that, over the concomitant period, the number of ice crystals detected by the ICE CAMERA dramatically decreased from ~700 to ~200, whatever the crystal ice habit size greater than 20 μm. In the presence on liquid particles, there is a strong Lidar raw signal together with a low depolarization signal as it is visible on Figure R8 around 100 m. Consequently, the reduced polarization detected in Fig. 14 reflects the presence of supercooled liquid water and not ice crystals.

We have inserted the Figure R4 in the new version of the manuscript and have rephrased the incriminated sentence into:

```
We  now  can  consider  the  time  evolution  of  the  number  of  ice
crystals  measured  by  the  ICE  CAMERA  integrated  over  one  hour
from  4  to  5  March  2013  above  the  Dome  C  station  on  Figure  15.
We  notice  that,  from  00:00  to  10:00  UTC  on  5  March,  the  number
of  ice  crystals  detected  by  the  ICE  CAMERA  dramatically
decreased  from  ~700  to  ~200,  whatever  the  crystal  ice  habit
size  greater  than  20  μm.  There  is  also  a  strong  Lidar  raw
signal  (not  shown)  together  with  a  low  Lidar  depolarization
signal  (Fig.  7)  around  100  m,  signature  of  liquid  particles.
Consequently,  the  reduced  polarization  detected  in  Fig.  14
reflects  the  presence  of  supercooled  liquid  water  and  not  the
presence  of  ice  crystals.
```

[Figure]

**Fig. 14:** Time evolution of the Depolarization ratio (%) from 4 to 5 March 2013 above the Dome C station as measured by the aerosol Lidar installed at Dome C.

[Figure]

*Figure R7: Time evolution of the number of ice crystals measured by the ICE CAMERA integrated over one hour from 4 to 5 March 2013 above the Dome C station.*

[Figure]

*Figure R8: Time evolution of the Lidar raw signal (top) and depolarization (bottom) measured on 5 March 2013 above the Dome C station.*

26) Lines 496-498 – What vertical structure are you referring to? There is considerable vertical structure in the depolarization across the time period 1000-2400 on March 5 and within the lower few hundred metres. You can't say with any certainty here that there was no precipitation of ice crystals. Even diamond dust is large enough to fall out unless there is sufficient vertical wind to maintain their suspension. Please revise.

→ Indeed, the reviewer is right. When considering the depolarization ratio measured on 5 March 2013 (Figure R9) in the lowermost troposphere (0-500 m), some traces of vertical structures are present around 18:00 UTC, signature of precipitation. In the literature, it is not easy to estimate the fall velocity of ice crystals depending whether we consider ice fog or diamond dust. Bürgesser et al. (2016) refer to small (ice fog) particles; whilst Böhm (1989) refer to large ice crystals (diamond dust). From Bürgesser et al. (2016), the fall velocity for columnar ice crystal with lengths ranging 20-160 µm is ranging 1-6 cm s$^{-1}$. From Böhm (1989), the terminal velocity for radiative assemblage of dentrites, unrimed plates, side planes, bullets and columns is ranging 0.35-0.75 m s$^{-1}$ for diameters ranging 0.2-1.0 mm and the terminal velocity of unrimed side planes with diameters ranging 0.4-1.0 mm is ranging 0.65-0.85 m s$^{-1}$. But it is beyond the scope of the present paper to discuss the fall velocity of the ice crystals.

[Figure]

*Figure R9: Time evolution of the Depolarization Ratio (%) measured on 5 March 2013 above the Dome C station from the surface to 500 m above the ground.*

Böhm, H. P.: A general equation for the terminal fall speed of solid hydrometeors. J. Atmos. Sci., 46, 2419-2427, 1989.

Bürgesser, R. E., Ávila, E. E. and Castellano, N. E.: Laboratory measurements of sedimentation velocity of columnar ice crystals. Q. J. R. Meteorol. Soc., 142, 1713-1720, 2016. doi:10.1002/qj.2766.

Note that when considering the depolarization ratio measured on 4-6 April 2011 (Figure R10) in the lowermost troposphere (0-500 m), some traces of vertical structures are also present, signature of precipitation. Consequently, we cannot mention, as we did L. 411 of the previous

version, "there is no trace of precipitation". Therefore, we have rephrased the sentence in the revised version of the manuscript.

```
The high depolarization ratio shows that the cloud is
constituted of ice crystals and, since there are no other
layers higher in the troposphere (as during the warm and wet
period), there is little trace of precipitation.
```

[Figure]

*Figure R10*: *Time evolution of the Depolarization Ratio (%) measured on 4-6 April 2011.*

27) Line 510 – Please revise to ". . . of ice crystals with a longer suspension time in the air."
→ Done.

28) Lines 520-523 – change "were presented" on line 522 to "is discussed". You are discussing the "impact".
→ No, the subject is not "the impact" but both "the impact" and "the high correlation". We modified the incriminated sentence into:

```
Both the impact of the origin of air masses on the short-term
variability of H₂O and temperature and the high correlation
coefficient (greater than 0.90) between water vapour and
temperature at Dome C over the entire year 2010 were presented
in Ricaud et al. (2012 and 2014c) based on 5-day back-
trajectory calculations.
```

29) Line 564 – Replace "If we consider the" with "The". Remove "as".
→ Done.

30) Line 565 – Remove the first "the".
→ Done.

31) Line 566 – Replace ", we obviously remark" with "show".
→ Done.

32) Line 569 – Remove "indeed".
→ Done.

33) Line 575 – Replace "slight" with "smaller".

→ Done.

34) Line 578 – "thick-cloud episodes"
→ Done.

35) Line 580 – ". . . the diamond dust episodes occurred during. . ."
→ Done. We have rephrased the incriminated sentence into:

```
Consequently, considering episodes 1 and 2, the thick-cloud
episodes observed during the warm and wet period above Dome C are
attributed to air masses with an oceanic origin whilst the diamond
dust/ice fog episodes occurred during the cold and dry period are
attributed to air masses with continental origins.
```

36) Line 584 – "Here, we attribute the tendencies of . . ."
→ Done.

37) Line 585 and 586 – Replace "into" by "among"
→ Done.

38) Section 5.3 – Please explain why this attribution was not be done for episode 1?
→ The budget computation is done during the model integration on a pre-defined area around Dome C. It requires a large-scale area to estimate the advection tendency but can be done only over a short period, as during episode 2 with the ARPEGE global-scale model. AROME is a meso-scale model and can provide this parameter only to within its domain (250x250 km) and for a long period (episode 1). So the advection tendency is not meaningful for AROME during episode 1.

39) Line 604 – remove "a". change "on" to "in"
→ Done.

40) Line 607 – what are "small precipitations"?
→ The term "weak precipitations" has been used instead of "small precipitations".

41) Line 609-610 – Does this sentence, which refers to dehydration of the PBL and includes "precipitation", contradict your statement at the end of section 4.4 that the diamond dust does not precipitate? See also above comments 26 and 27.
→ You are right. There are indeed traces of precipitation. See discussions 26.

42) Lines 600-610 – You refer to microphysics as the one of the factors influencing the water vapour budget. I know you are using the model with microphysics, but it is not clear from your discussion if the influence on the water budget is truly microphysics or just the presence of cloud. Please clarify in Section 5.3.
→ The negative tendency of Qv only means condensation so water vapour transforms to liquid or ice droplets. Clouds have only impact on the temperature budget through the radiation.

43) Line 650 – "Since both downward and upward longwave radiation are greater than. . ."
→ Done.

44) Line 665-666 – ". . . suggesting that the cloud consisted of smaller ice crystals that may remain suspended in the air longer."
→ Done.

In conclusion, 5 new references have been inserted in the revised manuscript:

Bazile, E., Couvreux, F., Le Moigne, P., and Genthon, C.: First Workshop on the GABLS-4 Intercomparison, GEWEX Newsletter, pp18-19, August 2015.

Girard, E. and Blanchet, J. P.: Microphysical parameterization of Arctic diamond dust, ice fog, and thin stratus for climate models. J. Atmos. Sci., 58, 1181-1198, 2001.

Gultepe, I., Kuhn, T., Pavolonis, M., Calvert, C., Gurka, J., Heymsfield, A. J., Liu, P. S. K., Zhou, B., Ware, R., Ferrier, B., Milbrandt, J. and Berstein, B.:Ice fog in Arctic during FRAM-ICE Fog project: Aviation and Nowcasting Applications, Bull. Am. Meteor. Soc., 95, 211-226, 2014.

Lazzara, M.: Diagnosing Antarctic fog, 5[th] International Conference on Fog, Fog Collection and Dew, Münster, Germany, 25-30 July 2010. Vol. 1, p. 150. (http://meetingorganizer.copernicus.org/FOGDEW2010/FOGDEW2010-150.pdf)

Vié, B., Pinty, J. P., Berthet, S., & Leriche, M. : LIMA (v1. 0): A quasi two-moment microphysical scheme driven by a multimodal population of cloud condensation and ice freezing nuclei. Geosci. Model Dev., 9, 567-586, 2016.

---

## Referee Report (RR1)

REV 1

Title: Genesis of diamond dust, ice fog, and thick cloud……..

Submitted by Ricaud et al

Date Feb 14 2017

Decision: Rejection/Resubmit

***General Comments:***

I am still not happy with content of this work, it is not much improved. I feel authors should focus on what they like to show and they should stick with their goals. Even the objectives are not clear in this work, and figures are not represented properly.  Organizing of the text is also poor. How can someone use the results of this work for studying Arctic clouds/fog? I will not go details here but important issues are provided below randomly.

***Issues:***

Genesis means what? Even they do not summarize the cases properly and talking about genesis????; what these numbers mean? Abstract is too long. Look at page 7; LN131-136;  I see they are not focused and clear. LN132 episode 1 talks about DD, IF episodes,  and LN135-136 genesis of DD, IF, Thick cloud…….

These clearly show this paper doesn't analyze the cases properly, and objectives are not given in the end of introduction but in the middle of the text (page 6). This is very awkward  writing style.

Ice Camera; hourly particles are analyzed, if heavy snow happens, crystals will cover the glass in a few seconds, if ice fog/DD will be covered in a few seconds also. What does really mean analyze the data hourly? To me nothing important having this data set, except some selected ice crystals. You say 5 micron resolution, are you sure?  Fig. R1 and R2 have scale of 1000 micron. I cant see particles less than 100-200 micron. What is going on here?.

Looking at the data you have, I don't see fog particles?

Fig. 9; a) IWC and b) PF (precip flux); when I see highest IWC, there is no precip? How can I trust this image/fig?

Also; 0.015 mm/day PF=0.0015 mm/day; this means no precipitation basically. How do you explain this?

Fig. 8; integrated over 1 hr? you have 1000/hr=(1000/3600) /(Lsec)~0.3/(sec), to me this is a very low number (certainly do not represent ice fog). What is the sampling area (page 10)? If I assume 10x10cm2, and 10  cm/sec fall speed, this makes about 0.2/L which is very low number, do you call this ice fog, DD, or snow crystals? I see no clear comparison or calculations of ice microphysical properties with other studies.

Fig. 6; this is what? Absolute humidity? Or cloud water content? Please show IWC and RH for the same case from MWR and lidar, and then model simulations.

I feel that authors are pushing their results desperately to be published; in fact, publishing these kinds of work should be careful performed. This manuscript should be rejected and resubmitted with more focused way and it should be designed properly.

I suggest authors should follow up the structure below

1) Set up clear goals
2) Better description of observations
3) Better comparisons of the results with others
4) Focus on only DD/Snow or IF, and present results that comparable with others, if not, explain why? Their characteristics are not the same
5) Provide insight to figures, IWC, Ni, precip flux etc.
6) Show simulation of model matching observed quantities.
7) Clear conclusions;
   a) Found that Ni is comparable …..
   b) DD Ni was ……
   c) NWP results were …..
   d) …….
   e) New sensor showed that…….

---

## Author Response (AR2)

**Manuscript Title:** *Genesis of Diamond Dust and Thick Cloud Episodes observed above Dome C, Antarctica* **by Ricaud et al.**

**Revised title:** *Genesis of Diamond Dust, Ice Fog and Thick Cloud Episodes observed and modelled above Dome C, Antarctica*

**RESPONSES TO THE REVIEWERS**

We would like to thank the reviewers for their insightful comments that were helpful in improving substantially the presentation and contents of the revised manuscript. We have addressed appropriately all issues raised by the reviewers. The reviewers' comments are repeated below in blue and our responses appear in black.

We have modified this sentence in the acknowledgments:

> We finally would like to thank the ==three== anonymous reviewers to their fruitful comments.

Changes have been highlighted in yellow in the revised manuscript.

**Anonymous Referee #1**

REV 1

Title: Genesis of diamond dust, ice fog, and thick cloud……..

Submitted by Ricaud et al

Date Feb 14 2017

Decision: Rejection/Resubmit

*General Comments:*

I am still not happy with content of this work, it is not much improved. I feel authors should focus on what they like to show and they should stick with their goals. Even the objectives are not clear in this work, and figures are not represented properly. Organizing of the text is also poor. How can someone use the results of this work for studying Arctic clouds/fog? I will not go details here but important issues are provided below randomly.

*Issues:*

1) Genesis means what? Even they do not summarize the cases properly and talking about genesis????; what these numbers mean? Abstract is too long. Look at page 7; LN131-136; I see they are not focused and clear. LN132 episode 1 talks about DD, IF episodes, and LN135-136 genesis of DD, IF, Thick cloud…….

These clearly show this paper doesn't analyze the cases properly, and objectives are not given in the end of introduction but in the middle of the text (page 6). This is very awkward writing style.

→ We thank the referee for the time to review the manuscript. The main concern of the referee relates to the organization and writing style of the manuscript. However, two out of three referees have no issue with the manuscript in its current form. We therefore prefer to keep the organization of the manuscript as is. Below we address the clarifications requested by the referee.

2) Ice Camera; hourly particles are analyzed, if heavy snow happens, crystals will cover the glass in a few seconds, if ice fog/DD will be covered in a few seconds also. What does really mean analyze the data hourly? To me nothing important having this data set, except some selected ice crystals. You say 5 micron resolution, are you sure? Fig. R1 and R2 have scale of 1000 micron. I cant see particles less than 100-200 micron. What is going on here?.

→ We recall again that measurements are performed in Antarctica and not in the Arctic. The precipitation rate in Concordia is so weak that the ICE-CAMERA plate is fully covered of precipitation only on very special occasions, approximately 10-15 days per year. Ice fog never covers the plate because it is electrically-heated and the particles sublimated after each photo.

The resolution is actually 5 microns (7-10 microns when the focusing device is not at its best). The maximum size on the plots presented in Figs R1 and R2 is indeed 1000 microns but there is no link with the image resolution. And, in any case, the picture is compressed for sending it via E-mail to Italy. Consequently, the overall quality of the Figure is worsened.

3) Looking at the data you have, I don't see fog particles?

→ See reply to the comments #6.

4) Fig. 9; a) IWC and b) PF (precip flux); when I see highest IWC, there is no precip? How can I trust this image/fig?

→ In the microphysics scheme inserted in ARPEGE, above a threshold value (based on a formulation), the cloud water (or ice) is transformed (auto conversion) into precipitation (solid or liquid). After, in the layer just below and during the same time step, the evaporation of precipitation is computed (only if the relative humidity is less than 100%). But, in general, the evaporation of precipitation is very small.

So, in Fig. 9, the model starts to create some IWC (Ice Water Content) and, in one layer (~3 km), the value becomes greater than the threshold value ($Qc\_crit$). Consequently, precipitation is generated in order to reduce IWC to $Qc\_crit$. The layer below is probably dryer and then precipitations are evaporated.

In our system, the threshold value for auto conversion depends on the environment but it is around $10^{-5}$.

5) Also; 0.015 mm/day PF=0.0015 mm/day; this means no precipitation basically. How do you explain this?

→ See reply to the comments #4. But we agree with the reviewer, the amount of precipitation is very small in the model. This is probably why precipitation disappears by evaporation. This only means that the model evaluates that some ice cloud droplets start to fall in precipitation because IWC > $Qc\_crit$ for some time steps.

6) Fig. 8; integrated over 1 hr? you have 1000/hr=(1000/3600) /(Lsec)~0.3/(sec), to me this is a very low number (certainly do not represent ice fog). What is the sampling area (page 10)? If I assume 10x10cm2, and 10 cm/sec fall speed, this makes about 0.2/L which is very low number, do you call this ice fog, DD, or snow crystals? I see no clear comparison or calculations of ice microphysical properties with other studies.

→ The picture is indeed taken every hour. Then the plate is heated for 15 minutes and the collection restarts. Because the collection time is approximately 45 minutes and is not reproducible, the data recorded from the ICE-CAMERA are quantitatively not reliable, but the size and the shape of deposited particles are.

The question relative to the nature of the particles deposited cannot be answered with our study for several reasons. The distinction between ice fog and diamond dust was asked to be reviewed by the referee in the previous version of the manuscript. Unfortunately, the paper Gultepe et al. (2013) and related literature only refer to arctic data. We have to recall that the precipitation rate in Concordia is much smaller than the rates observed in the Arctic, with concentrations of

less than 1000 part/l. Moreover, "arctic" ice-type at Concordia is probably undetected by the ICE-CAMERA because of the small size/deposition rate of the tiniest crystals. Finally, we cannot state with our own measurements and modelling studies whether the low ice precipitation episodes at Concordia are diamond dust or ice fog episodes. This is the reason why we have modified our conclusions together with the abstract and the title of the revised version of the manuscript.

7) Fig. 6; this is what? Absolute humidity? Or cloud water content? Please show IWC and RH for the same case from MWR and lidar, and then model simulations.

→ As already written in the Figure caption, Figure 6 shows the "Time evolution of **absolute humidity** from 15 March to 8 April 2011 above Dome C as measured by the HAMSTRAD radiometer from 0 to 5 km."

8) I feel that authors are pushing their results desperately to be published; in fact, publishing these kinds of work should be careful performed. This manuscript should be rejected and resubmitted with more focused way and it should be designed properly.

I suggest authors should follow up the structure below
1) Set up clear goals
2) Better description of observations
3) Better comparisons of the results with others
4) Focus on only DD/Snow or IF, and present results that comparable with others, if not, explain why? Their characteristics are not the same
5) Provide insight to figures, IWC, Ni, precip flux etc.
6) Show simulation of model matching observed quantities.
7) Clear conclusions;
a) Found that Ni is comparable …..
b) DD Ni was ……
c) NWP results were …..
d) …….
e) New sensor showed that…….

→ Please see our response to organization above (reply to the comments #1).